# Chaos of Learning Beyond Zero-sum and Coordination via Game Decompositions

**Yun Kuen Cheung**
Department of Computer Science
Royal Holloway University of London
Egham, UK
yunkuen.cheung@rhul.ac.uk

**Yixin Tao**
Department of Mathematics
London School of Economics
London, UK
Y.Tao16@lse.ac.uk

## Abstract

It is of primary interest for Machine Learning to understand how agents learn and interact dynamically in competitive environments and games (e.g. GANs). But this has been a difficult task, as irregular behaviors are commonly observed in such systems. This can be explained theoretically, for instance, by the works of Cheung & Piliouras (2019; 2020), which showed that in two-person zero-sum games, if agents employ one of the most well-known learning algorithms, Multiplicative Weights Update (MWU), then *Lyapunov chaos* occurs everywhere in the cumulative payoff space. In this paper, we study how persistent chaos can occur in the general normal-form game settings, where the agents might have the motivation to coordinate (which is not true for zero-sum games) and the number of agents can be arbitrary.

We characterize bimatrix games where MWU, its optimistic variant (OMWU) or Follow-the-Regularized-Leader (FTRL) algorithms are Lyapunov chaotic almost everywhere in the cumulative payoff space. Our characterizations are derived by extending the volume-expansion argument of Cheung & Piliouras via the canonical game decomposition into zero-sum and coordination components. Interestingly, the two components induce opposite volume-changing behaviors, so the overall behavior can be analyzed by comparing the *strengths* of the components against each other. The comparison is done via our new notion of "matrix domination" or via a linear program. For multi-player games, we present a *local equivalence of volume change* between general games and graphical games, which is used for volume and chaos analyses of MWU and OMWU in potential games.

## 1 Introduction

In Machine Learning (ML), it is of primary interest to understand how agents learn in competitive environments. This is more strongly propelled recently due to the success of Generative Adversarial Networks (GANs), which can be viewed as two neural-networks playing a zero-sum game. As such, Evolutionary Game Theory (EGT) (Hofbauer & Sigmund (1998); Sandholm (2010)), a decades-old area devoted to the study of adaptive (learning) behaviors of agents in competitive environments arising from Economics, Biology, and Physics, has drawn attention from the ML community. In contrast with the typical optimization (or no-regret) approach in ML, EGT provides a dynamical-systemic perspective to understand ML processes, which has already provided new insights into a number of ML-related problems. This perspective is particularly helpful in studying "learning in games", where irregular behaviors are commonly observed, but the ML community currently lacks of a rigorous method to analyze such systems. In this paper, we study *Lyapunov chaos*, a central notion that captures instability and unpredictability in dynamical systems. We characterize general normal games where popular learning algorithms exhibit chaotic behaviors.

*Lyapunov chaos* captures the *butterfly effect*: when the starting point of a dynamical system is slightly perturbed, the resulting trajectories and final outcomes diverge quickly; see Definition 1 for a formal definition. The perturbations correspond to round-off errors of numerical algorithms in ML (and Computer Science in general).[1] While significant efforts have been spent in analyzing and

---

[1] In games, such perturbations can also occur due to errors in measuring payoffs.

minimizing round-off effects of floating-point computations (Demmel (1997)), they are unavoidable in general[2]. As round-offs are inevitable, and the round-off schemes can vary from machine to machine due to various hardware and software factors, we surely want to avoid chaotic learning that does not fulfill our primary goals in building *predictable* and *reproducible* learning systems. Such issues are exemplified by a quote from Ali Rahimi's NIPS'2017 test-of-time award speech:

> *"Someone on another team changed the default rounding mode of some Tensor-flow internals from 'truncate toward zero' to 'round to even'. Our training broke, our error rate went from less than $25\%$ error to $\sim 99.97\%$ error."*

To avoid chaotic learning, we first need to understand how it can arise. This is the main motivation of our work. Recently, in the context of "learning in games", Cheung & Piliouras (2019; 2020) presented interesting theoretical analyses to show that in two-person zero-sum and graphical constant-sum games, if the agents employ Multiplicative Weights Update (MWU) or Follow-the-Regularized-Leader (FTRL) algorithms, then Lyapunov chaos occurs everywhere in the cumulative payoff space; the same result holds for the optimistic variant of MWU (OMWU) in coordination games.[3] While zero-sum and coordination games are interesting in their own rights, they are rather small subspaces within the family of general normal games. In this paper, we present techniques and tools for characterizing general games where MWU, FTRL, or OMWU are Lyapunov chaotic almost everywhere. Next, we give an overview of our contributions and a discussion of related work.

**Our Contributions.** To show the results about chaos mentioned above, Cheung & Piliouras (2019; 2020) used a classical technique in the study of dynamical systems called *volume analysis*. Volume analysis considers a set of starting points of positive Lebesgue measure, e.g. a ball centred at a point; volume is an alternative name for Lebesgue measure in this context. When this set of starting points evolves according to the rule of dynamical system, it becomes a new set with a different volume. Intuitively, volume is a measure of the range of possible outcomes, so the larger it is, the more unpredictable the system is. If the set's volume increases exponentially with time, then its diameter increases exponentially too, which implies Lyapunov chaos. This indicates that when players repeatedly play the games by employing the respective learning algorithms, a slight perturbation on the initiating condition can lead to a wide range of possible cumulative payoffs in the long run. This can be shown to imply instability in the mixed strategy space.

The technical starting point of Cheung & Piliouras is to show that when all agents in a bimatrix game $\mathbf{G}$ use MWU with step-size $\epsilon$, and when a set $S$ in the cumulative payoff space evolves for one time step, its volume change can be expressed as $\epsilon^2 \int_S C_{\mathbf{G}}(s)\, ds + \mathcal{O}(\epsilon^4)$, where $C_{\mathbf{G}}$ is a function that depends on the game $\mathbf{G}$, which we will define in Section 2. Clearly, the sign of $C_{\mathbf{G}}$ dictates the volume change behaviors for all small enough $\epsilon$. Cheung & Piliouras showed that $C_{\mathbf{G}}$ is a positive function when $\mathbf{G}$ is a two-person zero-sum game. For a large region in the cumulative payoff space, this implies the volume change per time step is $\Omega(\epsilon^2) \cdot \text{volume}(S)$, i.e. volume expands exponentially. They also showed that $C_{\mathbf{G}}$ is a negative function if $\mathbf{G}$ is a two-person coordination game.

To extend their volume expansion results to general bimatrix games, we first discover that $C_{\mathbf{G}}$ admits a clean decoupling w.r.t. the canonical decomposition of such games into zero-sum and coordination components (Basar & Ho (1974); Kalai & Kalai). Precisely, given any two-person general game $(\mathbf{A}, \mathbf{B})$, it can be written uniquely as a direct sum of a zero-sum game $(\mathbf{Z}, -\mathbf{Z})$ and a coordination game $(\mathbf{C}, \mathbf{C})$, where $\mathbf{Z} = (\mathbf{A} - \mathbf{B})/2$ and $\mathbf{C} = (\mathbf{A} + \mathbf{B})/2$. Interestingly, we find that $C_{\mathbf{G}}(\cdot) = C_{(\mathbf{Z}, -\mathbf{Z})}(\cdot) + C_{(\mathbf{C}, \mathbf{C})}(\cdot)$ in Lemma 6. Recall from the last paragraph that $C_{(\mathbf{Z}, -\mathbf{Z})}(\cdot)$ is always positive, while $C_{(\mathbf{C}, \mathbf{C})}(\cdot)$ is always negative. Thus, to see whether volume expansion occurs, it boils down to comparing the strengths of $C_{(\mathbf{Z}, -\mathbf{Z})}(\cdot)$ and $-C_{(\mathbf{C}, \mathbf{C})}(\cdot)$.

We also discover that the function $C_{\mathbf{G}}$ is invariant upon additions of *trivial matrices* to the game $\mathbf{G}$; see Definition 7 and Lemma 8. An immediate application of trivial matrices is for bimatrix potential games (Monderer & Shapley (1996)), for which we show it can be transformed to a coordination game via additions of trivial matrices. With the result in Cheung & Piliouras (2020), this immediately implies that OMWU in any bimatrix potential game is Lyapunov chaotic everywhere in the cumulative payoff space (Observation 10).

Based on the above discoveries, we identify two characterizations of bimatrix games where MWU and FTRL are Lyapunov chaotic almost everywhere (Theorems 15 and 17). As said before,

---

[2]For instance, in learning algorithms we often evaluate the function $e^x$, while $e$ is an irrational number.

[3]They also showed some other chaos results for various combinations of learning algorithms and games.

the key is to compare the strengths of the $C$-functions of the zero-sum and the coordination components. The comparison is done via our new notion of *matrix domination* (see Definition 11), and also a linear program (Eqn. (7)) which is designed to prune out the trivial-matrix projection and keep the remaining part minimal. Such family of games has positive Lebesgue measure in the bimatrix game space, so it is not confined to any proper game subspace[4]. This justifies the claim that the occurrences of chaos are not only circumstantial, but a rather substantial issue in learning in games. Analogous result holds for OMWU.

For games with any number of players (multi-player games), we use an observation in Cheung & Piliouras (2019), coupled with our new findings about bimatrix games discussed above, to present a new family of graphical games where MWU is Lyapunov chaotic almost everywhere (Theorem 18); the new family strictly includes all graphical constant-sum games. To facilitate volume analyses of learning in multi-player games, we establish their *local equivalence of volume change* with graphical games. Briefly, we show that $C_{\mathbf{G}}(\mathbf{p})$ for a general game $\mathbf{G}$ is the same as $C_{\mathbf{H}}(\mathbf{p})$ for some graphical game $\mathbf{H}$; $\mathbf{H}$ will depend on the point $\mathbf{p}$, that's why we say the equivalence is *local* (Theorem 19). This provides an intuitive procedure for understanding volume changes, as the volume change of learning in a graphical game is easier to compute. This is used to show that the volume-changing behaviors of MWU and OMWU are opposite to each other. We use these to analyze MWU and OMWU in multi-player potential games; in particular, we show that $C_{\mathbf{G}}(\mathbf{p})$ of a multi-player potential game $\mathbf{G}$ is identical to $C_{\mathbf{C}}(\mathbf{p})$ of a corresponding multi-player graphical coordination game, while $C_{\mathbf{C}}(\mathbf{p}) \leq 0$ for any $\mathbf{p}$ (Proposition 21).[5]

**Related Work.** MWU and its variant, such as FTRL and Optimistic MWU, play important roles in online learning. We recommend the texts of Cesa-Bianchi & Lugosi (2006) and Hart & Mas-Colell (2013) for a modern overview of online learning from Machine Learning or Economics perspectives.

Recently, there is a stream of works that examine how learning algorithms behave in games or min-max optimization from a dynamical-systemic perspective. This provides new insights into learning systems which could hardly be obtained with classical tools in ML. For instance, some learning systems are shown to be nearly-periodic under the notion of *Poincaré recurrence* (Piliouras & Shamma (2014); Mertikopoulos et al. (2018)). Using potential functions first proposed in EGT and other tools from mathematics, surprising behaviors of first-order methods in zero-sum games and min-max optimization were discovered (Daskalakis & Panageas (2018; 2019); Bailey & Piliouras (2018); Cheung (2018)). In Appendix A, we give an account on some further related work.

Empirical evidences of Lyapunov chaos of learning in games were reported by Sato et al. (2002) and Galla & Farmer (2013). *Li-Yorke chaos*, another classical chaos notion, was proved to occur in several learning-in-game systems (Palaiopanos et al. (2017); Chotibut et al. (2020)).

Volume analysis has long been a technique of interest in the study of population and game dynamics. It was discussed in a number of famous texts; see (Hofbauer & Sigmund, 1998, Section 11), (Fudenberg & Levine, 1998, Section 3) and (Sandholm, 2010, Chapter 9).

We use game decomposition in this paper, which is a natural and generic approach to extend compelling results for a specific family of games to more general games. Let $\mathcal{H}$ denote the specific family of games with some compelling properties. Given a general game, we seek to decompose it into a sum of its *projection* on $\mathcal{H}$, plus one or more *residue* components. If the residues are small, then it is plausible that those compelling properties extend (approximately). In seeking of games where learning is stable, game decompositions were used (Candogan et al. (2011; 2013a;b); Letcher et al. (2019)) with $\mathcal{H}$ being potential games.

## 2 PRELIMINARY

In this paper, every bold lower-case alphabet denotes a vector, every bold upper-case alphabet denotes a matrix or a game. When we say a "game", we always mean a normal-form game. Given $n$, let $\Delta^n$ denote the mixed strategy space of dimension $n-1$[6], i.e. $\{(z_1, z_2, \cdots, z_n) \mid \sum_{j=1}^{n} z_j = 1\}$.

**Normal-Form Games.** Let $N$ denote the number of players in a game. Let $S_i$ denote the strategy set of Player $i$, and $S := S_1 \times \cdots \times S_N$. Let $n_i = |S_i|$. $\mathbf{s} = (s_1, \cdots, s_N) \in S$ denotes a strategy

---

[4]The families of zero-sum and coordination games are proper subspaces of the bimatrix game space. Any proper subspace has Lebesgue measure zero.

[5]The theorem and propositions mentioned in this paragraph will be stated formally in Appendix C.

[6]The probability simplex over $n$ strategies is $(n-1)$-dimensional.

profile of all players, and $u_i(\mathbf{s})$ denotes the payoff to Player $i$ when each player picks $s_i$. A mixed strategy profile is denoted by $\mathbf{x} = (\mathbf{x}_1, \cdots, \mathbf{x}_N) \in \Delta^{n_1} \times \cdots \times \Delta^{n_N}$, and $u_i$ is extended to take mixed strategies as inputs via $u_i(\mathbf{x}) = \mathbb{E}_{\mathbf{s} \sim \mathbf{x}}[u_i(\mathbf{s})]$. We let $-(i_1, \cdots, i_g)$ denote the player set other than players $i_1, \cdots, i_g$. We also let

$$U^{i_1 i_2 \cdots i_g}_{j_1 j_2 \cdots j_g}(\mathbf{x}) = \mathbb{E}_{\mathbf{s}_{-(i_1, \cdots, i_g)} \sim \mathbf{x}_{-(i_1, \cdots, i_g)}} \left[ u_{i_1}(s_{i_1} = j_1, \cdots, s_{i_g} = j_g, \mathbf{s}_{-(i_1, \cdots, i_g)}) \right], \qquad (1)$$

which is the expected payoff to Player $i_1$ when: for $1 \leq f \leq g$, Player $i_f$ picks strategy $j_f$, while for each player $i \notin \{i_1, \cdots i_g\}$, she picks a strategy randomly following $\mathbf{x}_i$. We also use $U^{i_1 i_2 \cdots i_g}_{j_1 j_2 \cdots j_g}$ if $\mathbf{x}$ is clear from the context. We say a game is a *zero-sum game* if $\sum_i u_i(\mathbf{s}) = 0$ for all $\mathbf{s} \in S$, and we say a game is a *coordination game* if $u_i(\mathbf{s}) = u_k(\mathbf{s})$ for all Players $i$ and $k$ and for all $\mathbf{s} \in S$.

When $N = 2$, such games are called *bimatrix games*, for which we adopt the notations below. Let $(\mathbf{A}, \mathbf{B})$ denote a bimatrix game, where for any $j \in S_1$, $k \in S_2$, $A_{jk} := u_1(j, k)$, $B_{jk} := u_2(j, k)$. $\mathbf{x}$ and $\mathbf{y}$ denote mixed strategies of Players 1 and 2 respectively. A bimatrix game is a zero-sum game if $\mathbf{A} = -\mathbf{B}$; it is a coordination game if $\mathbf{A} = \mathbf{B}$. Note that $U^1_j = [\mathbf{A}\mathbf{y}]_j$, $U^2_k = [\mathbf{B}^\mathsf{T}\mathbf{x}]_k$, which we denote by $A_j, B_k$ respectively when $\mathbf{x}, \mathbf{y}$ are clear from context; $B_j, A_k$ are defined analogously.

**MWU, FTRL and OMWU in Games.** All three algorithms have a step-size $\epsilon$, and can be implemented as updating in the cumulative payoff (dual) space. In each round, the players' actions (mixed strategies) in the strategy space are functions of the cumulative payoff vectors to be defined below, and these actions are then used to determine the payoffs in the next round. For a player with $d$ strategies, let $\mathbf{p}^t \in \mathbb{R}^d$ denote her cumulative payoff vector at time $t$, and let $\mathbf{p}^0 \in \mathbb{R}^d$ denote the starting point chosen by the player. For MWU in a game, the update rule for Player $i$ is

$$p^{t+1}_j = p^t_j + \epsilon \cdot U^i_j(\mathbf{x}^t), \qquad (2)$$

where $U^i_j$ is the function defined in Eqn. (1), and $\mathbf{x}^t$ is the mixed strategy as below:

$$x^t_j = x_j(\mathbf{p}^t) = \exp(p^t_j) / (\textstyle\sum_{\ell \in S_i} \exp(p^t_\ell)) \qquad (3)$$

For OMWU in a game, the update rule for Player $i$ starts with $\mathbf{p}^1 = \mathbf{p}^0$, and for $t \geq 2$, $p^{t+1}_j = p^t_j + \epsilon \cdot \left[ 2U^i_j(\mathbf{x}^t) - U^i_j(\mathbf{x}^{t-1}) \right]$, where $\mathbf{x}^t$ is determined by Eqn. (3).

For FTRL in a game, the update rule for Player $i$ is same as Eqn. (2), but $\mathbf{x}^t$ is determined as below using a convex *regularizer function* $h_i : \Delta^d \to \mathbb{R}$: $\mathbf{x}^t = \arg\max_{\mathbf{x} \in \Delta^d} \{ \langle \mathbf{p}^t, \mathbf{x} \rangle - h_i(\mathbf{x}) \}$. As all the results for MWU can be directly generalized to FTRL as discussed in (Cheung & Piliouras, 2019, Appendix D), to keep our exposition simple, in the rest of this paper, we focus on MWU and OMWU, and their comparisons. For bimatrix game, we use $\mathbf{p}, \mathbf{q}$ to denote the cumulative payoff vectors of Players 1 and 2 respectively.

**Dynamical Systems, Lyapunov Chaos and Volume Analysis.** A learning-in-game system can be viewed as a discrete-time dynamical system, for which we present a simplified definition which suits our need. A discrete-time dynamical system in $\mathbb{R}^d$ is determined by a starting point $\mathbf{r}(0) \in \mathbb{R}^d$ and an update rule $\mathbf{r}(t+1) = f(\mathbf{r}(t))$, where $f : \mathbb{R}^d \to \mathbb{R}^d$ is a function.[7] The sequence $\mathbf{r}(0), \mathbf{r}(1), \mathbf{r}(2), \cdots$ is called a *trajectory* of the dynamical system. When $f$ is clear from the context, we let $\Phi : (\mathbb{N} \cup \{0\}) \times \mathbb{R}^d \to \mathbb{R}^d$ denote the function such that $\Phi(t, \mathbf{r})$ is the value of $\mathbf{r}(t)$ generated by the dynamical system with starting point set to $\mathbf{r}$. Given a set $\mathcal{U} \subset \mathbb{R}^d$, we let $\Phi(t, \mathcal{U}) = \{ \Phi(t, \mathbf{r}) | \mathbf{r} \in \mathcal{U} \}$. Let $\mathcal{B}(\mathbf{r}, z)$ denote the open ball with center $\mathbf{r}$ and radius $z$.

There are several similar but not identical definitions of Lyapunov chaos, all capturing the *butterfly effect*: when the starting point is slightly perturbed, the resulting trajectories diverge quickly. We use the following definition, which was also used by Cheung & Piliouras (2019; 2020) implicitly. Intuitively, a system is Lyapunov chaotic in an open set $\mathcal{O} \subset \mathbb{R}^d$ if for any $\mathbf{r} \in \mathcal{O}$ and any open ball $B$ around $\mathbf{r}$, as long as $\Phi(t, B)$ remains inside $\mathcal{O}$, there exists $\mathbf{r}' \in B$ such that $\|\Phi(t, \mathbf{r}') - \Phi(t, \mathbf{r})\|$ grows exponentially with $t$. Lyapunov exponent in the definition below is a measure of how fast the exponential growth is; the larger it is, the more unpredictable the dynamical system is.

---

[7] Rigorously, OMWU in game is not a dynamical system, as the update to $\mathbf{p}(t + 1)$ depends on both $\mathbf{p}(t), \mathbf{p}(t - 1)$. But there is a function $f$ such that $\mathbf{p}(t + 1) \approx f(\mathbf{p}(t))$, while the volume-changing behavior is not really affected (Cheung & Piliouras (2020)).

**Definition 1.** *A dynamical system is* Lyapunov chaotic[8] *in an open set $\mathcal{O} \subset \mathbb{R}^d$ if there exists a constant $\lambda > 0$ and a Lyapunov exponent $\gamma \equiv \gamma(\mathcal{O}) > 0$, such that for any $\mathbf{r} \in \mathcal{O}$, for any sufficiently small $\delta > 0$ and for all $t$ satisfying $0 \le t < \min\{\tau \mid \tau \ge 0, \ \Phi(\tau, \mathcal{B}(\mathbf{r}, \delta)) \not\subset \mathcal{O}\}$,*

$$\sup_{\mathbf{r}' \in \mathcal{B}(\mathbf{r}, \delta)} \|\Phi(t, \mathbf{r}') - \Phi(t, \mathbf{r})\| \ \ge \ \lambda \cdot \delta \cdot \exp(\gamma t).$$

**Definition 2.** *A dynamical system is* Lyapunov chaotic everywhere *if it is Lyapunov chaotic in any bounded open subset of $\mathbb{R}^d$.*

In the above definitions, all norms and radii are Euclidean norms. For capturing round-off errors in computer algorithms and ML systems, it is more natural to use $\ell_\infty$-norm with $\delta$ be the maximum round-off error per step, say $\sim 10^{-16}$ when IEEE 754 binary64 (standard double) is used.

When $\mathcal{O}$ is a small set, it is easy to determine whether a dynamical system is Lyapunov chaotic in $\mathcal{O}$, since the dynamic can be locally approximated by a linear dynamical system, where the eigenvalues of the local Jacobian characterizes chaotic behaviors. But when $\mathcal{O}$ is large, determining whether Lyapunov chaos occurs is difficult in general. Cheung & Piliouras (2019) found that volume analysis can be useful in this regard, based on the following simple observation.

**Proposition 3.** *In $\mathbb{R}^d$, if a set $\mathcal{U}$ has volume at least $v$, then its radius w.r.t. any point $\mathbf{r} \in \mathcal{U}$ is at least $v^{1/d}/2$. Thus, if the volume of $\Phi(t, \mathcal{U})$ of some dynamical system is $\Omega(\exp(\gamma t))$ for some $\gamma > 0$, then the radius of $\Phi(t, \mathcal{U})$ w.r.t. any point $\mathbf{r} \in \Phi(t, \mathcal{U})$ is $\Omega(\exp(\frac{\gamma}{d} \cdot t))$.*

Cheung and Piliouras showed Lemma 4 below, which, for bimatrix games, reduces volume analysis to analyzing the sign of the function $C_{(\mathbf{A}, \mathbf{B})}(\mathbf{p}, \mathbf{q})$ defined in Eqn. (4) below; the sign also determines the local volume-changing behavior around the point $(\mathbf{p}, \mathbf{q})$ when MWU is used. [9] Based on Proposition 3 that converts volume expansion to radius expansion, the sign can be used to determine if the dynamical system is Lyapunov chaotic.

Let $A_j = \sum_{k'} A_{jk'} y_{k'} = \nabla_{\mathbf{x}_j}[\mathbf{x}^\mathsf{T} \mathbf{A} \mathbf{y}]$ and $A_k = \sum_{j'} x_{j'} A_{j'k} = \nabla_{\mathbf{y}_k}[\mathbf{x}^\mathsf{T} \mathbf{A} \mathbf{y}]$; and, similarly, $B_j = \sum_{k'} B_{jk'} y_{k'} = \nabla_{\mathbf{x}_j}[\mathbf{x}^\mathsf{T} \mathbf{B} \mathbf{y}]$ and $B_k = \sum_{j'} x_{j'} B_{j'k} = \nabla_{\mathbf{y}_k}[\mathbf{x}^\mathsf{T} \mathbf{B} \mathbf{y}]$. Then,

$$C_{(\mathbf{A}, \mathbf{B})}(\mathbf{p}, \mathbf{q}) = -\mathbb{E}_{\mathbf{x}, \mathbf{y}}[(A_{jk} - A_j - A_k)(B_{jk} - B_j - B_k)] + \mathbb{E}_{\mathbf{x}, \mathbf{y}}[A_{jk}] \cdot \mathbb{E}_{\mathbf{x}, \mathbf{y}}[B_{jk}]. \quad (4)$$

Note that here $\mathbf{x}$ and $\mathbf{y}$ are the shorthand for $\mathbf{x}(\mathbf{p})$ and $\mathbf{y}(\mathbf{q})$, which are the mixed strategies (i.e. probability distributions over strategies) of Players 1 and 2 respectively, as computed via Eqn. (3). Also, $\mathbb{E}_{\mathbf{x}, \mathbf{y}}[f(j, k)] = \mathbb{E}_{(j,k) \sim (\mathbf{x}(\mathbf{p}), \mathbf{y}(\mathbf{q}))}[f(j, k)]$ is the expected value of $f(j, k)$ when the strategies $j$ and $k$ are randomly chosen according to the distrubutions $\mathbf{x}(\mathbf{p})$ and $\mathbf{y}(\mathbf{q})$ respectively.

For multi-player game $\mathbf{G}$, the analogous function $C_{\mathbf{G}}(\cdot)$ is given below; the $U$ quantities were defined in Eqn. (1). Lemma 4 is adapted from Cheung & Piliouras (2019) for games with any number of players. Derivation of Eqn. (5) uses the Jacobian of the corresponding dynamical system and integration by substitution; see Appendix D.

$$C_{\mathbf{G}}(\mathbf{p}_1, \cdots, \mathbf{p}_N) = -\sum_{i \in [N], \ j \in S_i} \ \sum_{k > i, \ \ell \in S_k} x_{ij} x_{k\ell} \left(U_{\ell j}^{ki} - U_\ell^k\right)\left(U_{j\ell}^{ik} - U_j^i\right). \quad (5)$$

**Lemma 4.** *Suppose $\mathcal{O}$ is a set in the cumulative payoff space $\mathbb{R}^d$ where $d = n_1 + \cdots + n_N$, and*

$$\bar{c}_{\mathbf{G}}(\mathcal{O}) := \inf_{(\mathbf{p}_1, \cdots, \mathbf{p}_N) \in \mathcal{O}} C_{\mathbf{G}}(\mathbf{p}_1, \cdots, \mathbf{p}_N) \ > \ 0. \quad (6)$$

*Then for the dynamical system in which MWU with any sufficiently small step-size $\epsilon$ is employed to play the game $\mathbf{G}$, it is Lyapunov chaotic in $\mathcal{O}$ with Lyapunov exponent $\bar{c}_{\mathbf{G}}(\mathcal{O}) \cdot \epsilon^2/2d$.*

*If MWU is replaced by OMWU, then the same result holds by replacing the condition (6) with $\bar{c}_{\mathbf{G}}(\mathcal{O}) := \inf_{(\mathbf{p}_1, \cdots, \mathbf{p}_N) \in \mathcal{O}}[-C_{\mathbf{G}}(\mathbf{p}_1, \cdots, \mathbf{p}_N)] > 0$.*

---

[8]We note that many linear dynamical systems admits simple closed-form solutions (e.g. $dx/dt = x$ has solution $x(t) = x(0) \cdot e^t$), but they are considered Lyapunov chaotic under this definition. This might not match with the intuitive meaning of "chaos" to many people. However, for non-linear dynamical systems which mostly do not admit closed-form solutions (including all systems we study), Lyapunov chaos is well-received as a notion that captures unpredictability.

[9]They showed that, in the cumulative payoff space, when the set $S^t$ is evolved to $S^{t+1}$ after one MWU step, then $\texttt{volume}(S^{t+1}) = \texttt{volume}(S^t) + \epsilon^2 \int_{(\mathbf{p}, \mathbf{q}) \in S^t} C_{(\mathbf{A}, \mathbf{B})}(\mathbf{p}, \mathbf{q}) \, d\mathbf{p} \, d\mathbf{q} + \mathcal{O}(\epsilon^4)$. Thus, if $C_{(\mathbf{A}, \mathbf{B})}(\mathbf{p}, \mathbf{q}) > 0$, then the volume is increasing, which indicates diverging trajectories, i.e. chaos.

Note that if we start from a Nash equilibrium in the strategy space, MWU and OMWU will stay at the equilibrium. However, if this equilibrium $(\mathbf{x}^*, \mathbf{y}^*)$ satisfies the conditions in Corollary 5 below, there are points arbitrarily close to the equilibrium that keep moving away from the equilibrium (if the region $\{(\mathbf{x}', \mathbf{y}') = (\mathbf{x}(\mathbf{p}), \mathbf{y}(\mathbf{q})) | (\mathbf{p}, \mathbf{q}) \in \mathcal{O}\}$ is large in the strategy simplex $\Delta^{n_1} \times \Delta^{n_2}$).

**Corollary 5** (Adapted from (Cheung & Piliouras, 2020, Theorem 5)). *Let $(\mathbf{x}^*, \mathbf{y}^*)$ be a point in the interior of the strategy space. Suppose that there exists $(\mathbf{p}, \mathbf{q})$ in the cumulative payoff space, such that $\mathbf{x}^* = \mathbf{x}(\mathbf{p}^*)$ and $\mathbf{y}^* = \mathbf{y}(\mathbf{q}^*)$. Furthermore, suppose $C_{(\mathbf{A},\mathbf{B})}(\mathbf{p}^*, \mathbf{q}^*) > 0$, and $(\mathbf{p}, \mathbf{q}) \in \mathcal{O}$ where $\mathcal{O}$ is the set described in Lemma 4. Then there are strategy points arbitrarily close to $(\mathbf{x}^*, \mathbf{y}^*)$ such that MWU in the game $(\mathbf{A}, \mathbf{B})$ eventually leaves the corresponding strategy set of $\mathcal{O}$, i.e. $\{(\mathbf{x}', \mathbf{y}') = (\mathbf{x}(\mathbf{p}), \mathbf{y}(\mathbf{q})) | (\mathbf{p}, \mathbf{q}) \in \mathcal{O}\}$.*

We give the intuitions behind the proof of Corollary 5. Suppose the contrary, i.e. for any open neighbourhood of $(\mathbf{p}^*, \mathbf{q}^*)$, its flow never escapes from $\mathcal{O}$. Then there are two contradicting facts. First, the volume of the flow expands at least exponentially with time. Second, by Eqn. (2), each $p_j^t$ grows at most linearly with $t$ (since $|U_j^i| \leq \max_{j,k}\{|A_{jk}|, |B_{jk}|\}$), and thus the volume of the flow can only expand at most polynomially with time.

When the game is zero-sum, i.e., $\mathbf{B} = -\mathbf{A}$, hence $C_{(\mathbf{A},\mathbf{B})}(\mathbf{p}, \mathbf{q}) = \mathbb{E}_{\mathbf{x},\mathbf{y}}\left[(A_{jk} - A_j - A_k)^2\right] - \mathbb{E}_{\mathbf{x},\mathbf{y}}\left[(A_{jk})\right]^2$. Since $\mathbb{E}_{\mathbf{x},\mathbf{y}}[A_{jk}] = \mathbb{E}_{\mathbf{x},\mathbf{y}}[A_j] = \mathbb{E}_{\mathbf{x},\mathbf{y}}[A_k]$ and hence $\mathbb{E}_{\mathbf{x},\mathbf{y}}[A_{jk} - A_j - A_k] = -\mathbb{E}_{\mathbf{x},\mathbf{y}}[A_{jk}]$, $C_{(\mathbf{A},\mathbf{B})}(\mathbf{p}, \mathbf{q})$ is indeed the variance of the random variable $A_{jk} - A_j - A_k$, and thus is non-negative. By Eqn. (4), we have $C_{(\mathbf{A},\mathbf{B})}(\mathbf{p}, \mathbf{q}) = -C_{(\mathbf{A},-\mathbf{B})}(\mathbf{p}, \mathbf{q})$. Thus, for any coordination game $(\mathbf{A}, \mathbf{A})$, we have $C_{(\mathbf{A},\mathbf{A})}(\mathbf{p}, \mathbf{q}) = -C_{(\mathbf{A},-\mathbf{A})}(\mathbf{p}, \mathbf{q}) \leq 0$.

## 3 BIMATRIX GAMES

In this section, we focus on general bimatrix games $(\mathbf{A}, \mathbf{B})$. In Section 3.1, we present two tools for analyzing $C_{(\mathbf{A},\mathbf{B})}(\cdot)$, and then we provide an example to show how to use these tools. In Section 3.2, we present two characterizations such that the dynamics are Lyapunov chaotic almost everywhere.

### 3.1 TOOLS FOR ANALYZING BIMATRIX GAME

**First Tool: Canonical Decomposition for Bimatrix Games.** For every bimatrix game $(\mathbf{A}, \mathbf{B})$, it admits a canonical decomposition (Basar & Ho (1974); Kalai & Kalai) into the sum of a zero-sum game $(\mathbf{Z}, -\mathbf{Z})$ and a coordination game $(\mathbf{C}, \mathbf{C})$, where $\mathbf{Z} = \frac{1}{2}(\mathbf{A} - \mathbf{B})$ and $\mathbf{C} = \frac{1}{2}(\mathbf{A} + \mathbf{B})$, i.e.

$$(\mathbf{A}, \mathbf{B}) = (\mathbf{Z}, -\mathbf{Z}) + (\mathbf{C}, \mathbf{C}).$$

We call $(\mathbf{Z}, -\mathbf{Z})$ the zero-sum part of the game $(\mathbf{A}, \mathbf{B})$, and $(\mathbf{C}, \mathbf{C})$ the coordination part of the game. Our first result shows that the function $C(\cdot)$ can be decomposed neatly into the two parts too.

**Lemma 6.** *For any bimatrix game $(\mathbf{A}, \mathbf{B})$,*

$$C_{(\mathbf{A},\mathbf{B})}(\mathbf{p}, \mathbf{q}) \equiv C_{(\mathbf{Z},-\mathbf{Z})}(\mathbf{p}, \mathbf{q}) + C_{(\mathbf{C},\mathbf{C})}(\mathbf{p}, \mathbf{q}).$$

*Proof.* We use Eqn. (4) to expand the following:

$$4 \cdot C_{(\mathbf{Z},-\mathbf{Z})}(\mathbf{p}, \mathbf{q}) + 4 \cdot C_{(\mathbf{C},\mathbf{C})}(\mathbf{p}, \mathbf{q})$$
$$= \mathbb{E}\left[(A_{jk} - B_{jk} - A_j + B_j - A_k + B_k)^2\right] - \mathbb{E}[A_{jk} - B_{jk}]^2$$
$$\qquad - \mathbb{E}\left[(A_{jk} + B_{jk} - A_j - B_j - A_k - B_k)^2\right] + \mathbb{E}[A_{jk} + B_{jk}]^2$$
$$= \mathbb{E}\left[(A_{jk} - B_{jk} - A_j + B_j - A_k + B_k)^2 - (A_{jk} + B_{jk} - A_j - B_j - A_k - B_k)^2\right]$$
$$\qquad - (\mathbb{E}[A_{jk}] - \mathbb{E}[B_{jk}])^2 + (\mathbb{E}[A_{jk}] + \mathbb{E}[B_{jk}])^2$$
$$= \mathbb{E}[4(-B_{jk} + B_j + B_k)(A_{jk} - A_j - A_k)] + 4 \cdot \mathbb{E}[A_{jk}] \cdot \mathbb{E}[B_{jk}] \quad = \quad 4 \cdot C_{(\mathbf{A},\mathbf{B})}(\mathbf{p}, \mathbf{q}). \quad \square$$

By the end of Section 2, we discussed that $C_{(\mathbf{Z},-\mathbf{Z})}(\mathbf{p}, \mathbf{q})$ is always non-negative and $C_{(\mathbf{C},\mathbf{C})}(\mathbf{p}, \mathbf{q})$ is always non-positive. By the above lemma, we can analyze the volume-changing behavior of a bimatrix game $(\mathbf{A}, \mathbf{B})$ by looking at its zero-sum and coordination parts independently. One simple intuition is that if the coordination (resp. zero-sum) part is small, then the volume-changing behavior of $(\mathbf{A}, \mathbf{B})$ is closer to the behavior of the zero-sum (resp. coordination) part. We realize this intuition quantitatively in the next subsection.

**Second Tool: Trivial matrix.** Trivial matrices are matrices which do not affect the volume-changing behavior, as depicted in Lemma 8 below.

**Definition 7** (Trivial Matrix). $\mathbf{T} \in \mathbb{R}^{n \times m}$ *is a* trivial matrix *if there exists real numbers* $u_1, u_2, \cdots, u_n$ *and* $v_1, v_2, \cdots, v_m$ *such that* $T_{jk} = u_j + v_k$ *for all* $j \in [n], k \in [m]$.

**Lemma 8.** *For any two trivial matrices* $\mathbf{T}^1, \mathbf{T}^2$*, for any two matrices* $\mathbf{A}, \mathbf{B} \in \mathbb{R}^{n \times m}$*,*

$$C_{(\mathbf{A}, \mathbf{B})}(\mathbf{p}, \mathbf{q}) \equiv C_{(\mathbf{A} + \mathbf{T}^1, \mathbf{B} + \mathbf{T}^2)}(\mathbf{p}, \mathbf{q}).$$

One immediate application of this lemma is for two player potential games.

**Definition 9.** *A game* $\mathbf{G}$ *is a potential game if there exists a potential function* $\mathcal{P} : S \to \mathbb{R}$ *such that for any Player* $i$ *and any strategy profile* $\mathbf{s} \in S$, $\mathcal{P}(s_i, \mathbf{s}_{-i}) - \mathcal{P}(s_i', \mathbf{s}_{-i}) = u_i(s_i, \mathbf{s}_{-i}) - u_i(s_{i'}, \mathbf{s}_{-i})$.

For the potential game, we have the following observation:

**Observation 10.** *For any bimatrix potential game* $(\mathbf{A}, \mathbf{B})$*, there is a coordination game* $(\mathbf{P}, \mathbf{P})$ *such that* $\mathbf{A} - \mathbf{P}$, $\mathbf{B} - \mathbf{P}$ *are trivial matrices.* $\mathbf{P}$ *is the matrix representation of the potential function* $\mathcal{P}$*.*

This observation immediately implies that the volume-changing behavior of a potential game is equivalent to that of a corresponding coordination game.

We give a concrete example to show how these tools help us to analyze the $C_{(\mathbf{A}, \mathbf{B})}(\cdot)$.

**A Simple Example.** We will show how to use our tools to demonstrate $C(\cdot) \geq 0$ everywhere for the following game. In the example, each player has three strategies. The payoff bimatrix $(\mathbf{A}, \mathbf{B})$ is given below. The first number gives the payoff of the row player, who chooses a strategy from $\{a, b, c\}$; the second number gives the payoff of the column player, who chooses a strategy from $\{1, 2, 3\}$. We first use our *first tool* to decompose this game into zero-sum part $(\mathbf{Z}, -\mathbf{Z})$ and

|              | Strategy 1 | Strategy 2 | Strategy 3 |
|--------------|------------|------------|------------|
| Strategy $a$ | $(4, 4)$   | $(12, -4)$ | $(-6, 10)$ |
| Strategy $b$ | $(-8, 8)$  | $(0, 0)$   | $(12, -4)$ |
| Strategy $c$ | $(14, -2)$ | $(-8, 8)$  | $(4, 4)$   |

coordination part $(\mathbf{C}, \mathbf{C})$, where $\mathbf{Z} = \begin{bmatrix} 0 & 8 & -8 \\ -8 & 0 & 8 \\ 8 & 0 & 0 \end{bmatrix}$ and $\mathbf{C} = \begin{bmatrix} 4 & 4 & 2 \\ 0 & 0 & 4 \\ 6 & 0 & 4 \end{bmatrix}$. At this point, we still cannot easily figure out which one is larger between $C_{(\mathbf{Z}, -\mathbf{Z})}(\cdot)$ and $C_{(\mathbf{C}, -\mathbf{C})}(\cdot)$. However, we can further decompose the coordination part by the *second tool*: $\mathbf{C} = \begin{bmatrix} 4 & 2 & 4 \\ 2 & 0 & 2 \\ 4 & 2 & 4 \end{bmatrix} + \begin{bmatrix} 0 & 2 & -2 \\ -2 & 0 & 2 \\ 2 & -2 & 0 \end{bmatrix}$, where the first matrix on the RHS is a trivial matrix (using notations in Definition 7, $u = v^{\mathsf{T}} = [2, 0, 2]$). It's easy to see the second matrix on the RHS is $\frac{1}{4}\mathbf{Z}$. Then by Lemmas 6 and 8, and the definition of the function $C$, for any point $(\mathbf{p}, \mathbf{q})$ in the cumulative payoff space,

$$C_{(\mathbf{A}, \mathbf{B})}(\mathbf{p}, \mathbf{q}) = C_{(\mathbf{Z}, -\mathbf{Z})}(\mathbf{p}, \mathbf{q}) + C_{(\frac{1}{4}\mathbf{Z}, \frac{1}{4}\mathbf{Z})}(\mathbf{p}, \mathbf{q}) = \left(1 - (1/4)^2\right) \cdot C_{(\mathbf{Z}, -\mathbf{Z})}(\mathbf{p}, \mathbf{q}) \geq 0.$$

## 3.2 RESULTS FOR BIMATRIX GAMES

In this subsection, we identify several characterizations for general bimatrix games in which we have chaotic behavior with MWU dynamic in a following set $S^\delta$ in the cumulative payoff space $\mathbb{R}^{n_1 + n_2}$. Note that when $\delta$ is tiny, its strategy correspondence covers almost the entirety of the strategy simplex $\Delta^{n_1} \times \Delta^{n_2}$, thus we informally say that if the dynamical system is Lyapunov chaotic in $S^\delta$ for a tiny $\delta$, then it is *Lyapunov chaotic almost everywhere*.

$$S^\delta = \{(\mathbf{p}, \mathbf{q}) \,|\, \forall j \in S_1, k \in S_2, \ x_j(\mathbf{p}) \geq \delta \ \wedge \ y_k(\mathbf{q}) \geq \delta\}.$$

In order to show chaotic behavior of MWU in a specific bimatrix game $(\mathbf{A}, \mathbf{B})$, it is sufficient to show $C_{(\mathbf{A}, \mathbf{B})}(\mathbf{p}, \mathbf{q})$ is strictly positive in the region $S^\delta$, due to Lemma 4. In the previous subsection, we show that for each game $(\mathbf{A}, \mathbf{B})$, it can be decomposed into a zero-sum part $(\mathbf{Z}, -\mathbf{Z})$ and a coordination part $(\mathbf{C}, \mathbf{C})$. Furthermore, $C_{(\mathbf{A}, \mathbf{B})}(\mathbf{p}, \mathbf{q}) = C_{(\mathbf{Z}, -\mathbf{Z})}(\mathbf{p}, \mathbf{q}) + C_{(\mathbf{C}, \mathbf{C})}(\mathbf{p}, \mathbf{q})$. We also raise an intuition that if the zero-sum part is small, then the volume behavior in the game $(\mathbf{A}, \mathbf{B})$ will be similar that in the coordination part; conversely, if the coordination part is small, then the volume behavior will be similar to the zero-sum part. However, we have not yet presented a way to compare the largeness of the two parts. This is what we do here.

### 3.2.1 First Characterization: Matrix Domination

The first characterization we identify is matrix domination. In this part, we show that under certain conditions, the zero-sum part is always no less than the coordination part, i.e. $C_{(\mathbf{Z},-\mathbf{Z})}(\mathbf{p},\mathbf{q}) \geq -C_{(\mathbf{C},\mathbf{C})}(\mathbf{p},\mathbf{q})$ for all $(\mathbf{p},\mathbf{q})$. This directly implies $C_{(\mathbf{A},\mathbf{B})}(\mathbf{p},\mathbf{q})$ will be non-negative in the whole cumulative payoff space. Interestingly, the condition we identify is both necessary and sufficient. Similar result can also be achieved in the case that coordination part is always no less than the zero-sum part. We first introduce the definition of the matrix domination.

**Definition 11.** *We say matrix $\mathbf{K}$ dominates matrix $\mathbf{L}$ if they are of the same dimension, and for any row indices $j, j'$ and column indices $k, k'$,*

$$\left|\mathbf{K}_{jk} + \mathbf{K}_{j'k'} - \mathbf{K}_{jk'} - \mathbf{K}_{j'k}\right| \geq \left|\mathbf{L}_{jk} + \mathbf{L}_{j'k'} - \mathbf{L}_{jk'} - \mathbf{L}_{j'k}\right|.$$

Note that the domination induces a partial order on all matrices: if $\mathbf{K}$ dominates $\mathbf{L}$ and $\mathbf{L}$ dominates $\mathbf{M}$, then $\mathbf{K}$ dominates $\mathbf{M}$. The theorem below gives the necessary and sufficient condition.

**Theorem 12.** *$C_{(\mathbf{A},\mathbf{B})}(\mathbf{p},\mathbf{q})$ is non-negative for all $(\mathbf{p},\mathbf{q})$ if and only if matrix of the zero-sum part $\mathbf{Z}$ dominates the matrix of the coordination part $\mathbf{C}$.*

The above theorem is based on the following crucial observation.

**Observation 13.** *For any matrix $\mathbf{Z}$,*

$$C_{(\mathbf{Z},-\mathbf{Z})}(\mathbf{p},\mathbf{q}) = \frac{1}{4} \sum_{\substack{j,j' \in S_1 \\ k,k' \in S_2}} x_j(\mathbf{p}) \cdot y_k(\mathbf{q}) \cdot x_{j'}(\mathbf{p}) \cdot y_{k'}(\mathbf{q}) \cdot \left(Z_{jk} + Z_{j'k'} - Z_{jk'} - Z_{j'k}\right)^2.$$

Matrix domination only implies $C_{(\mathbf{A},\mathbf{B})}(\mathbf{p},\mathbf{q})$ is non-negative. In order to have $C_{(\mathbf{A},\mathbf{B})}(\mathbf{p},\mathbf{q})$ to be strictly positive in the set $S$, we need $\theta$-domination.

**Definition 14.** *We say matrix $\mathbf{K}$ $\theta$-dominates ($\theta > 0$) matrix $\mathbf{L}$ if $\mathbf{K}$ dominates $\mathbf{L}$, and there exist $j$, $j'$, $k$, $k'$ such that $\left|\mathbf{K}_{jk} + \mathbf{K}_{j'k'} - \mathbf{K}_{jk'} - \mathbf{K}_{j'k}\right| \geq \left|\mathbf{L}_{jk} + \mathbf{L}_{j'k'} - \mathbf{L}_{jk'} - \mathbf{L}_{j'k}\right| + \theta$.*

The following theorem holds due to Lemma 4.

**Theorem 15.** *For any general bimatrix game $(\mathbf{A},\mathbf{B})$ which is decomposed into zero-sum part $(\mathbf{Z},-\mathbf{Z})$ and coordination part $(\mathbf{C},\mathbf{C})$, if $\mathbf{Z}$ $\theta$-dominates $\mathbf{C}$, then MWU with any sufficiently small step-size $\epsilon$ in the game $(\mathbf{A},\mathbf{B})$ is Lyapunov chaotic in $S^\delta$ with Lyapunov exponent $\frac{\theta^2 \delta^4}{2(n_1 + n_2)}\epsilon^2$.*

Note that in Definition 14, $\mathbf{K}$ $\theta$-dominates $\mathbf{L}$ if a finite number of inequalities are satisfied. In the context of Theorem 15, it is easy to see that there are quite many games $(\mathbf{A},\mathbf{B})$, such that $\mathbf{Z}$ $\theta$-dominates $\mathbf{C}$ with all those inequalities *strictly* satisfied. Thus, there exists an open neighbourhood around these games such that every game in the neighbourhood has its zero-sum part $\theta$-dominates its coordination part. This shows that such family of games has positive Lebesgue measure.

### 3.2.2 Second Characterization: Linear Program

Note that matrix domination is not always true. In some scenarios, the zero-sum matrix might not dominate the coordination matrix. Yet, it is still possible that $C_{(\mathbf{A},\mathbf{B})}(\mathbf{p},\mathbf{q})$ is strictly positive in the region $S^\delta$, when every entry in the coordination matrix is *small*.

Precisely, for a general bimatrix game $(\mathbf{A},\mathbf{B})$, if its coordination part $(\mathbf{C},\mathbf{C})$ is small in the sense that the absolute values of all entries in $\mathbf{C}$ are smaller than some constant $r$, then we can bound $C_{(\mathbf{C},-\mathbf{C})}(\cdot)$ by $\mathcal{O}(r^2)$. This is not the only case where we can bound $C_{(\mathbf{C},-\mathbf{C})}(\cdot)$ by a small term. Even if the entries in matrix $\mathbf{C}$ are large, we can use trivial matrices to reduce them without affecting $C_{(\mathbf{C},-\mathbf{C})}(\cdot)$. This is done via a linear programming approach described below.

Given a matrix $\mathbf{K}$, let $r(\mathbf{K})$ be the optimal value of following linear program:

$$\min_{r \geq 0, \mathbf{g}, \mathbf{h}} \quad r \text{ such that } \forall j, k, \quad -r \leq K_{jk} - g_j - h_k \leq r. \tag{7}$$

Note that $\{g_j + h_k\}_{j,k}$ is a trivial matrix. Let $\mathbf{K}' = \mathbf{K} - \{g_j + h_k\}_{j,k}$. By Lemma 8, $C_{(\mathbf{K},-\mathbf{K})}(\cdot) = C_{(\mathbf{K}',-\mathbf{K}')}(\cdot)$. The following lemma shows that the value of $C_{(\mathbf{K},-\mathbf{K})}(\cdot)$ is closely related to $r(\mathbf{K})$.

**Lemma 16.** *For any $(\mathbf{p},\mathbf{q})$ in $S^\delta = \{(\mathbf{p},\mathbf{q}) \,|\, \forall j, k, \; x_j(\mathbf{p}) \geq \delta \text{ and } y_k(\mathbf{q}) \geq \delta\}$, $(r(\mathbf{K}) \cdot \delta)^2 \leq C_{(\mathbf{K},-\mathbf{K})}(\mathbf{p},\mathbf{q}) \leq r(\mathbf{K})^2$.*

Then the theorem below follows by applying Lemma 16 with Lemma 4.

**Theorem 17.** *For any general bimatrix game* $(\mathbf{A}, \mathbf{B})$ *which is decomposed into zero-sum part* $(\mathbf{Z}, -\mathbf{Z})$ *and coordination part* $(\mathbf{C}, \mathbf{C})$, *if there exists* $\theta > 0$ *such that* $(r(\mathbf{Z}) \cdot \delta)^2 \geq (r(\mathbf{C}))^2 + (\theta \delta^2)^2$, *then MWU with any sufficiently small step-size* $\epsilon$ *in the game* $(\mathbf{A}, \mathbf{B})$ *is Lyapunov chaotic in* $S^\delta$ *with Lyapunov exponent* $\frac{\theta^2 \delta^4}{2(n_1 + n_2)} \epsilon^2$.

Intuitively, $r(\mathbf{Z})$ is a distance measure from the zero-sum game $(\mathbf{Z}, -\mathbf{Z})$ to the trivial game space; analogously, $r(\mathbf{C})$ is a distance measure from the coordination game $(\mathbf{C}, \mathbf{C})$ to the trivial game space. Theorem 17 shows that if the coordination part is much closer to the trivial game space than the zero-sum part, then MWU in this game is Lyapunov chaotic in $S^\delta$.

## 4 EXPERIMENT

To illuminate that volume expansion occurs when MWU is employed in game with small coordination part, we simulate MWU in the *reduced* payoff space[10] in a game which is the sum of zero-sum game $\left( \left[ \begin{smallmatrix} 1 & 0 \\ 0 & 1 \end{smallmatrix} \right], \left[ \begin{smallmatrix} -1 & 0 \\ 0 & -1 \end{smallmatrix} \right] \right)$ and coordination game $\left( \left[ \begin{smallmatrix} -0.05 & 0.03 \\ 0.03 & -0.05 \end{smallmatrix} \right], \left[ \begin{smallmatrix} -0.05 & 0.03 \\ 0.03 & -0.05 \end{smallmatrix} \right] \right)$.

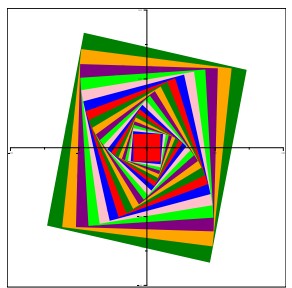

In the strategy space, $\mathbf{x}^* = \mathbf{y}^* = (0.5, 0.5)$ is the unique Nash equilibrium of the game. In the reduced dual space, the origin corresponds to the equilibrium. We pick a square of side length 0.004 around the origin as the set of starting points (the small red square in the middle of Figure 1). As these starting points are evolved via MWU with step-size 0.02, we take snapshots after every 1900 time steps, which are shown with colors blue, pink, lime, purple, orange and green (then the colors repeat) respectively. As shown in the figure, the volume (which is area in two-dimensional space) increases, and its shape changes from a square to a parallelogram.

Figure 1: Volume expansion of MWU in the bimatrix game $\left( \left[ \begin{smallmatrix} 0.95 & 0.03 \\ 0.03 & 0.95 \end{smallmatrix} \right], \left[ \begin{smallmatrix} -1.05 & 0.03 \\ 0.03 & -1.05 \end{smallmatrix} \right] \right)$

## 5 CONCLUSION AND FUTURE WORKS

In this paper, we analyze the volume-changing behavior of several well-known learning algorithms (MWU, OMWU, FTRL) on general bimatrix games and multi-player games, which leads to a Lyapunov chaos analysis. For bimatrix games, we do this by decomposing a game into the zero-sum part and the coordination part. This decomposition turns the volume analysis into comparing the strengths of volume-expansion (zero-sum part) and volume-contraction (coordination part) of the MWU dynamics. The comparison of strengths is made via the notion of matrix domination and the use of a linear program. For multi-player games, by the local equivalence, we show that the volume-changing behavior of MWU and OMWU are opposite to each other even in multi-player games. We also show, for a general multi-player potential game, the key function $C_{\mathbf{G}}$ is equal to a corresponding multi-player coordination game, which implies it is not positive.

Studying learning in matrix (normal-form) games, which are among one of the most classical game models, is a good theoretical starting point. Matrix games admit mathematically amenable analyses, as demonstrated in our work and many previous works. For future works, we are immensely interested in chaos analyses on settings that are more relevant to applications in ML, e.g. general GANs and differential games. We believe the techniques we use (volume analysis, game decomposition, etc.) can be applicable.

ACKNOWLEDGMENTS

We thank several anonymous reviewers for their suggestions, which help to improve the readability of this paper from its earlier version. Yixin Tao acknowledges NSF grant CCF-1527568, CCF-1909538 and ERC Starting Grant ScaleOpt-757481. Yun Kuen Cheung acknowledges Singapore NRF 2018 Fellowship NRF-NRFF2018-07.

---

[10]Reduced payoff space is a projection of the original payoff space. In the reduced payoff space, the horizontal axis is $p_1 - p_2$, while the vertical axis is $q_1 - q_2$. The original cumulative payoff space is four-dimensional, for which it is difficult to give graphical illumination. This is why we use the reduced space instead.

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

## A    FURTHER RELATED WORK

In the study of no-regret learning (e.g. Littlestone & Warmuth (1994); Freund & Schapire (1995)), a vast literature concerns general or even adversarial settings, in which the online arrivals of payoff values come with no pattern or even from an adversary. More recently, settings where the online payoffs are more well-behaved, under the term of "predictable sequence" coined by Rakhlin & Sridharan (2013b), have been studied. These settings include game dynamics, as the online payoffs are determined by the mixed strategy choices of the players, while these choices are updated gradually and somewhat predictably. For these settings, online learning algorithms that perform particularly well, e.g. achieving regret bound below the canonical $\mathcal{O}(\sqrt{T})$ limit, are designed and studied (Hazan & Kale (2010); Chiang et al. (2012); Syrgkanis et al. (2015)). For instance, Nesterov's excessive gap technique and optimistic mirror descent are found to achieve near-optimal regret $\mathcal{O}(\log T)$ in zero-sum games (Daskalakis et al. (2015); Rakhlin & Sridharan (2013a)), and thus the empirical average of the learning sequence converges to Nash equilibrium of the game (see Freund & Schapire (1996) for an explanation). OMWU (with time-varying step-sizes), and more generally optimistic variant of FTRL (Rakhlin & Sridharan (2013b)), are some canonical examples of such online learning algorithms.

Recently, there is a stream of work that examines how learning algorithms behave in games or min-max optimization from a dynamical-systemic perspective. Replicator dynamics (RD; the continuous-time analogue of MWU) and continuous-time FTRL are found to achieve optimal regret in general settings (Mertikopoulos et al. (2018)). Furthermore, RD in zero-sum games or graphical constant-sum games admits a constant of motion and preserves volume; these two properties are used to show that such dynamical systems are near-periodic (Piliouras & Shamma (2014); Mertikopoulos et al. (2018); Boone & Piliouras (2019); Vlatakis-Gkaragkounis et al. (2019); Perolat et al. (2020)), captured rigorously under the notion of *Poincaré recurrence* (Poincaré (1890); Barreira (2006)). However, when MWU, the forward Euler discretization of RD, is used in discrete-time setting in zero-sum games, the near-periodicity is destroyed totally; indeed, the system will never visit the same point (or its tiny neighbourhood) twice, converge to the boundary of the strategy simplex, and fluctuate there irregularly (Bailey & Piliouras (2018); Cheung (2018)). In contrast, (discrete-time) OMWU in zero-sum game is shown to converge to Nash equilibrium (Daskalakis & Panageas (2019)); yet, in the more general setting of min-max optimization, it was found that Optimistic Gradient Descent Ascent (OGDA) can have limit points other than (local) min-max solutions (Daskalakis & Panageas (2018)).

## B    PROOFS IN SECTION 3

*Proof of Lemma 8.* First, observe that it suffices to prove that the lemma holds when $\mathbf{T}^1$ is a trivial matrix and $\mathbf{T}^2$ is the zero matrix. Then the lemma holds for any trivial matrices $\mathbf{T}^1, \mathbf{T}^2$ due to symmetry: $C_{(\mathbf{A},\mathbf{B})}(\mathbf{p},\mathbf{q}) = C_{(\mathbf{A}+\mathbf{T}^1,\mathbf{B})}(\mathbf{p},\mathbf{q}) = C_{(\mathbf{A}+\mathbf{T}^1,\mathbf{B}+\mathbf{T}^2)}(\mathbf{p},\mathbf{q})$.

Due to the definition of trivial matrix, we can write $T^1_{jk} = u_j + v_k$. Then

$$C_{(\mathbf{A}+\mathbf{T}^1,\mathbf{B})}(\mathbf{p},\mathbf{q}) - C_{(\mathbf{A},\mathbf{B})}(\mathbf{p},\mathbf{q})$$

$$= -\mathbb{E}\left[\left(A_{jk} + u_j + v_k - A_j - u_j - \sum_{\ell \in S_2} v_\ell y_\ell - A_k - v_k - \sum_{\ell \in S_1} u_\ell x_\ell\right)(B_{jk} - B_j - B_k)\right]$$

$$\quad + \mathbb{E}\left[A_{jk} + u_j + v_k\right] \cdot \mathbb{E}\left[B_{jk}\right] + \mathbb{E}\left[(A_{jk} - A_j - A_k)(B_{jk} - B_j - B_k)\right] - \mathbb{E}\left[A_{jk}\right] \cdot \mathbb{E}\left[B_{jk}\right]$$

$$= -\mathbb{E}\left[\left(-\sum_{\ell \in S_2} v_\ell y_\ell - \sum_{\ell \in S_1} u_\ell x_\ell\right)(B_{jk} - B_j - B_k)\right] + \mathbb{E}\left[u_j + v_k\right] \cdot \mathbb{E}\left[B_{jk}\right]$$

$$= \mathbb{E}\left[v_k + u_j\right] \cdot \mathbb{E}\left[B_{jk} - B_j - B_k\right] + \mathbb{E}\left[u_j + v_k\right] \cdot \mathbb{E}\left[B_{jk}\right].$$

By recalling that $\mathbb{E}\left[B_{jk} - B_j - B_k\right] = -\mathbb{E}\left[B_{jk}\right]$, we have $C_{(\mathbf{A}+\mathbf{T}^1,\mathbf{B})}(\mathbf{p},\mathbf{q}) - C_{(\mathbf{A},\mathbf{B})}(\mathbf{p},\mathbf{q}) = 0$. □

*Proof of Observation 10.* Let $\mathcal{P}_{jk}$ be the potential value of a potential game when Player 1 plays strategy $j$ and Player 2 plays strategy $k$. Then according to the definition the potential function, for any $j_1$, $j_2$ and $k$,

$$A_{j_1 k} - A_{j_2 k} = \mathcal{P}_{j_1 k} - \mathcal{P}_{j_2 k}.$$

In particular, for any $j, k$, $A_{jk} = \mathcal{P}_{jk} + A_{1k} - \mathcal{P}_{1k}$. This implies that there exists $\mathbf{v}$ such that $A_{jk} = \mathcal{P}_{jk} + v_k$ for any $j$ and $k$.

Similarly, there exists $\mathbf{u}$ such that $B_{jk} = \mathcal{P}_{jk} + u_j$ for any $j$ and $k$. This implies that any two-player potential games are coordination games plus trivial matrices. $\qquad\square$

*Proof of Theorem 12.* We first prove that if $\mathbf{Z}$ dominates $\mathbf{C}$, then $C_{(\mathbf{A},\mathbf{B})}(\mathbf{p}, \mathbf{q})$ is always non-negative. By Observation 13,

$$
\begin{aligned}
C_{(\mathbf{A},\mathbf{B})}(\mathbf{p}, \mathbf{q}) &= C_{(\mathbf{Z},-\mathbf{Z})}(\mathbf{p}, \mathbf{q}) + C_{(\mathbf{C},\mathbf{C})}(\mathbf{p}, \mathbf{q}) \\
&= C_{(\mathbf{Z},-\mathbf{Z})}(\mathbf{p}, \mathbf{q}) - C_{(\mathbf{C},-\mathbf{C})}(\mathbf{p}, \mathbf{q}) \\
&= \frac{1}{4} \sum_{j,j',k,k'} x_j(\mathbf{p}) y_k(\mathbf{q}) x_{j'}(\mathbf{p}) y_{k'}(\mathbf{q}) \cdot \\
&\qquad \left( (Z_{jk} + Z_{j'k'} - Z_{jk'} - Z_{j'k})^2 - (C_{jk} + C_{j'k'} - C_{jk'} - C_{j'k})^2 \right) \\
&\geq 0.
\end{aligned}
$$

In contrast, if $\mathbf{Z}$ does not dominate $\mathbf{C}$, then there exist $\hat{j}$, $\hat{j}'$, $\hat{k}$, $\hat{k}'$ and $\delta > 0$ such that

$$
\left( C_{\hat{j}\hat{k}} + C_{\hat{j}'\hat{k}'} - C_{\hat{j}\hat{k}'} - C_{\hat{j}'\hat{k}} \right)^2 \geq \left( Z_{\hat{j}\hat{k}} + Z_{\hat{j}'\hat{k}'} - Z_{\hat{j}\hat{k}'} - Z_{\hat{j}'\hat{k}} \right)^2 + \delta.
$$

For each $\eta > 0$, we construct $\mathbf{p}$ and $\mathbf{q}$ such that $x_{\hat{j}}(\mathbf{p}) = x_{\hat{j}'}(\mathbf{p}) = y_{\hat{k}}(\mathbf{q}) = y_{\hat{k}'}(\mathbf{q}) = \frac{1-\eta}{2}$. Furthermore, we let $\Upsilon$ denote the maximum absolute value of all entries in matrices $\mathbf{A}$ and $\mathbf{B}$. Then, for all $j$ and $k$, $|Z_{jk}| \leq \Upsilon$ and $|C_{jk}| \leq \Upsilon$. Therefore,

$$
\begin{aligned}
C_{(\mathbf{A},\mathbf{B})}(\mathbf{p}, \mathbf{q}) &= C_{(\mathbf{Z},-\mathbf{Z})}(\mathbf{p}, \mathbf{q}) + C_{(\mathbf{C},\mathbf{C})}(\mathbf{p}, \mathbf{q}) \\
&= C_{(\mathbf{Z},-\mathbf{Z})}(\mathbf{p}, \mathbf{q}) - C_{(\mathbf{C},-\mathbf{C})}(\mathbf{p}, \mathbf{q}) \\
&= \frac{1}{4} \sum_{j,j',k,k'} x_j(\mathbf{p}) y_k(\mathbf{q}) x_{j'}(\mathbf{p}) y_{k'}(\mathbf{q}) \cdot \\
&\qquad \left( (Z_{jk} + Z_{j'k'} - Z_{jk'} - Z_{j'k})^2 - (C_{jk} + C_{j'k'} - C_{jk'} - C_{j'k})^2 \right) \\
&\leq -\delta \left( \frac{1-\eta}{2} \right)^4 + |S_1|^2 \cdot |S_2|^2 \cdot \eta \cdot 16\Upsilon^2.
\end{aligned}
$$

The last inequality holds as $(C_{jk} + C_{j'k'} - C_{jk'} - C_{j'k})^2 - (Z_{jk} + Z_{j'k'} - Z_{jk'} - Z_{j'k})^2 \leq 16\Upsilon^2$. The value of $-\delta \left( \frac{1-\eta}{2} \right)^4 + |S_1|^2 \cdot |S_2|^2 \cdot \eta \cdot 16\Upsilon^2$ will be negative if we pick a small enough $\eta$. $\qquad\square$

*Proof of Observation 13.* Consider a random process, where $j, j' \in S_1$ are randomly picked according to distribution $\mathbf{x}(\mathbf{p})$, and $k, k' \in S_2$ are randomly picked according to distribution $\mathbf{y}(\mathbf{q})$. Then the RHS of Observation 13 can be expressed as $\frac{1}{4} \cdot \mathbb{E}\left[ (Z_{jk} + Z_{j'k'} - Z_{jk'} - Z_{j'k})^2 \right]$.

Then we expand the squared term in the expectation. Observing the symmetries within the expansion, we immediately have

$$
\frac{1}{4} \cdot \mathbb{E}\left[ (Z_{jk} + Z_{j'k'} - Z_{jk'} - Z_{j'k})^2 \right] = \mathbb{E}\left[ (Z_{jk})^2 \right] - \mathbb{E}\left[ Z_{jk} Z_{jk'} \right] - \mathbb{E}\left[ Z_{jk} Z_{j'k} \right] + \mathbb{E}\left[ Z_{jk} Z_{j'k'} \right].
$$

Let $Z_j := [\mathbf{Z}\mathbf{y}]_j$ and $Z_k = [\mathbf{Z}^\mathsf{T}\mathbf{x}]_k$. Then we have

$$
\mathbb{E}\left[ Z_{jk} Z_{jk'} \right] = \sum_{j,k} x_j y_k Z_{jk} \sum_{k'} y_{k'} Z_{jk'} = \sum_j x_j [\mathbf{Z}\mathbf{y}]_j \sum_k y_k Z_{jk} = \sum_j x_j (Z_j)^2 = \mathbb{E}\left[ (Z_j)^2 \right].
$$

Similarly, $\mathbb{E}\left[ Z_{jk} Z_{j'k} \right] = \mathbb{E}\left[ (Z_k)^2 \right]$. Lastly, $\mathbb{E}\left[ Z_{jk} Z_{j'k'} \right] = \mathbb{E}\left[ Z_{jk} \right]^2$. Thus, the RHS of Observation 13 is simplified to

$$
\mathbb{E}\left[ (Z_{jk})^2 \right] - \mathbb{E}\left[ (Z_j)^2 \right] - \mathbb{E}\left[ (Z_k)^2 \right] + \mathbb{E}\left[ Z_{jk} \right]^2.
$$

We complete the proof by noting that from the definition of $C_{(\mathbf{Z},-\mathbf{Z})}(\cdot)$ in Eqn. (4), $C_{(\mathbf{Z},-\mathbf{Z})}(\cdot)$ can be rewritten as

$$\mathbb{E}\left[(Z_{jk})^2\right] - \mathbb{E}\left[Z_j Z_{jk}\right] - \mathbb{E}\left[Z_k Z_{jk}\right] + \mathbb{E}\left[Z_{jk}\right]^2,$$

while $\mathbb{E}\left[Z_j Z_{jk}\right] = \sum_j x_j Z_j \sum_k y_k Z_{jk} = \sum_j x_j Z_j Z_j = \mathbb{E}\left[(Z_j)^2\right]$, and similarly $\mathbb{E}\left[Z_j Z_{jk}\right] = \mathbb{E}\left[(Z_k)^2\right]$. $\qquad\square$

*Proof of Theorem 15.* By Lemma 4, it suffices to prove $\bar{c}_{(\mathbf{A},\mathbf{B})}(S^\delta) = \inf_{(\mathbf{p},\mathbf{q})\in S^\delta} C_{(\mathbf{A},\mathbf{B})}(\mathbf{p},\mathbf{q}) \geq \theta^2 \delta^4$.

This holds because the matrix $\mathbf{Z}$ $\theta$-dominates $\mathbf{C}$, which implies there exist $j$, $j'$, $k$, and $k'$ such that

$$(\mathbf{Z}_{jk} + \mathbf{Z}_{j'k'} - \mathbf{Z}_{jk'} - \mathbf{Z}_{j'k})^2 \geq (\mathbf{C}_{jk} + \mathbf{C}_{j'k'} - \mathbf{C}_{jk'} - \mathbf{C}_{j'k})^2 + \theta^2.$$

By applying Observation 13, $C_{(\mathbf{Z},-\mathbf{Z})}(\mathbf{p},\mathbf{q}) \geq C_{(\mathbf{C},-\mathbf{C})}(\mathbf{p},\mathbf{q}) + \theta^2\delta^4$, because for $(\mathbf{p},\mathbf{q}) \in S^\delta$, every $x_j(\mathbf{p}), y_k(\mathbf{q}), x_{j'}(\mathbf{p}), y_{k'}(\mathbf{q})$ is at least $\delta$. By noting that $C_{(\mathbf{C},\mathbf{C})}(\mathbf{p},\mathbf{q}) = -C_{(\mathbf{C},-\mathbf{C})}(\mathbf{p},\mathbf{q})$ and $C_{(\mathbf{A},\mathbf{B})}(\mathbf{p},\mathbf{q}) = C_{(\mathbf{Z},-\mathbf{Z})}(\mathbf{p},\mathbf{q}) + C_{(\mathbf{C},\mathbf{C})}(\mathbf{p},\mathbf{q})$, the result follows. $\qquad\square$

*Proof of Lemma 16.* A key observation is the following equality:

$$C_{(\mathbf{K},-\mathbf{K})}(\mathbf{p},\mathbf{q}) = \min_{\mathbf{g},\mathbf{h}} F(\mathbf{g},\mathbf{h}), \text{ where } F(\mathbf{g},\mathbf{h}) = \sum_{j,k} x_j(\mathbf{p}) \cdot y_k(\mathbf{q}) \cdot (K_{jk} - g_j - h_k)^2. \quad (8)$$

Recall the notations $K_j = \sum_k y_k K_{jk}$ and $K_k = \sum_j x_j K_{jk}$, and we let $e := \sum_{j,k} x_j y_k K_{jk} \equiv \mathbb{E}[K_{jk}]$. The equality (8) holds due to the following observations: (i) $F(\mathbf{g},\mathbf{h})$ is a smooth convex function of its variables, thus all minimum points have the same function value; (ii) if $\frac{\partial F}{\partial g_j}$ and $\frac{\partial F}{\partial h_k}$ are all zero at some point $(\mathbf{g},\mathbf{h})$, then the point is a minimal point of $F$; (iii) $C_{(\mathbf{K},-\mathbf{K})}(\mathbf{p},\mathbf{q})$ is the variance of the random variable $K_{jk} - K_j - K_k$ (see the end of Section 2), and thus by a definition of variance,

$$\begin{aligned}
C_{(\mathbf{K},-\mathbf{K})}(\mathbf{p},\mathbf{q}) &= \mathbb{E}\left[(K_{jk} - K_j - K_k - \mathbb{E}[K_{jk} - K_j - K_k])^2\right] \\
&= \mathbb{E}\left[(K_{jk} - K_j - K_k + e)^2\right] \quad (\text{since } \mathbb{E}[K_{jk} - K_j - K_k] = -\mathbb{E}[K_{jk}] = -e) \\
&= \sum_{j,k} x_j y_k (K_{jk} - (K_j - e) - K_k)^2 = F(\mathbf{g}^\#,\mathbf{h}^\#),
\end{aligned}$$

where $g_j^\# = K_j - e$ and $h_k^\# = K_k$; and (iv) at $(\mathbf{g}^\#,\mathbf{h}^\#)$, the partial derivatives stated in (ii) are all zero.

With this observation and comparing this with the definition of $r(\mathbf{Z})$, it's easy to figure out that $C_{(\mathbf{K},-\mathbf{K})}(\mathbf{p},\mathbf{q}) \leq r(\mathbf{K})^2$.

To see $C_{(\mathbf{K},-\mathbf{K})}(\mathbf{p},\mathbf{q}) \geq (r(\mathbf{K}) \cdot \delta)^2$, we first let $\mathbf{g}^*$ and $\mathbf{h}^*$ to be the optimal choice of $\mathbf{g}$ and $\mathbf{h}$ in $C_{(\mathbf{K},-\mathbf{K})}(\mathbf{p},\mathbf{q}) = \min_{\mathbf{g},\mathbf{h}} \sum_{j,k} x_j(\mathbf{p}) \cdot y_k(\mathbf{q}) \cdot (K_{jk} - g_j - h_k)^2$. Due to the specification of the linear program (7), we have [11]

$$2 \cdot r(\mathbf{K}) \leq \max_{j,k}\{K_{jk} - g_j^* - h_k^*\} - \min_{j,k}\{K_{jk} - g_j^* - h_k^*\}.$$

Therefore,

$$\max\left\{\left(\max_{j,k}\{K_{jk} - g_j^* - h_k^*\}\right)^2, \left(\min_{j,k}\{K_{jk} - g_j^* - h_k^*\}\right)^2\right\} \geq r(\mathbf{K})^2.$$

This immediately implies that $C_{(\mathbf{K},-\mathbf{K})}(\mathbf{p},\mathbf{q}) \geq (r(\mathbf{K}) \cdot \delta)^2$. $\qquad\square$

---

[11] If this is not true, we can let $\mathbf{g}$ and $\mathbf{h}$ in $r(\mathbf{Z})$ to be $\mathbf{g}_j = \mathbf{g}_j^* - \frac{\max_{j,k}\{K_{jk} - g_j^* - h_k^*\} + \min_{j,k}\{K_{jk} - g_j^* - h_k^*\}}{2}$ and $\mathbf{h}_k = \mathbf{h}_k^*$. Then we can achieve $r(\mathbf{K}) = \frac{\max_{j,k}\{K_{jk} - g_j^* - h_k^*\} - \min_{j,k}\{K_{jk} - g_j^* - h_k^*\}}{2}$ which make $r(\mathbf{K})$ smaller.

## C  MULTI-PLAYER GAMES

Computing volume change of learning algorithm in multi-player game is slightly more involved than the two-player case. We present a local equivalence formula of volume change between normal-form and graphical games. This provides an intuitive procedure for understanding volume changes. Proposition 20 shows that in multi-player game, the volume-changing behaviors of MWU and OMWU are again *opposite* to each other (which was shown for bimatrix game in Cheung & Piliouras (2020)).

**Graphical Games.** A graphical game Kearns et al. (2001) is a special type of $N$-player game where the payoffs can be compactly represented. In a graphical game $\mathbf{H}$, for each pair of players $i, k$, there is an *edge-game* which is a bimatrix game between the two players, denoted by $(\mathbf{H}^{i,k}, (\mathbf{H}^{k,i})^{\mathsf{T}})$, where $\mathbf{H}^{i,k} \in \mathbb{R}^{n_i \times n_k}$ is the payoff matrix that denotes the payoffs to Player $i$. Then the payoff to Player $i$ at strategy profile $\mathbf{s} = (s_1, s_2, \cdots, s_N)$ is the sum of payoffs to Player $i$ in all her edge-games, i.e. $u_i(\mathbf{s}) = \sum_{k \neq i} H^{i,k}_{s_i, s_k}$. As is standard, this payoff function is extended via expectation when the inputs are mixed strategies.

Here, we first use an observation from Cheung & Piliouras (2019) to construct a family of multi-player graphical games where MWU is Lyapunov chaotic in $S^{N,\delta} := \{(\mathbf{p}_1, \cdots, \mathbf{p}_N) | \forall i \in [N], j \in S_i, \ x_{ij}(\mathbf{p}_i) \geq \delta\}$. It was observed that the function $C_{\mathbf{H}}(\mathbf{p})$ defined in Eqn. (5) is the sum of $C_{(\mathbf{H}^{i,k}, (\mathbf{H}^{k,i})^{\mathsf{T}})}(\mathbf{p}_i, \mathbf{p}_k)$ of all pairs of Players $i < k$. This observation yields Theorem 18.

**Theorem 18.** *Let $\mathcal{G}^{\uparrow}$ denote the family of bimatrix games which satisfy the condition either in Theorem 15 or in Theorem 17. In an $N$-player graphical game where each edge-game is drawn from $\mathcal{G}^{\uparrow}$, if all players are employing MWU with a sufficiently small step-size $\epsilon$, then the dynamical system is Lyapunov chaotic in $S^{N,\delta}$ with Lyapunov exponent $N(N-1)\theta^2 \delta^4 \epsilon^2 / 4 \sum_{i=1}^{N} n_i$.*

**Local Equivalence of General Games and Graphical Games.** Next, we present a theorem which connects the value of $C_{\mathbf{G}}(\mathbf{p})$ of a general game to $C_{\mathbf{H}}(\mathbf{p})$, where $\mathbf{H}$ is a graphical game.

**Theorem 19.** *Given an $N$-player normal-form game $\mathbf{G}$ and any point $\mathbf{p}$ in the cumulative payoff space, the value of $C_{\mathbf{G}}(\mathbf{p})$ is the same as $C_{\mathbf{H}}(\mathbf{p})$, where $\mathbf{H}$ is a graphical game specified as follows: for each pair of Players $i, k$ and $j \in S_i, \ell \in S_k$, the payoff to Player $i$ in her edge-game with Player $k$ when Player $i$ picks $j$ and Player $k$ picks $\ell$ is $H^{ik}_{j\ell} := U^{ik}_{j\ell}$, where $U^{ik}_{j\ell}$ is defined in Eqn. (1).*

This theorem shows that for any game $\mathbf{G}$, the value of $C_{\mathbf{G}}(\mathbf{p})$ is the same as in a particular graphical game, where each pair of players, $(i, k)$ play a bimatrix game whose utility is exactly the utility of the original game $\mathbf{G}$, but taking the expectation on the randomness of the other players' strategies. If the original game $\mathbf{G}$ is a graphical game, then in the graphical game $H^{ik}_{j\ell} = U^{ik}_{j\ell} + c_{-i,-k}$, where $c_{-i,-k}$ is a parameter which does not depend on Players $i$ and $k$.

Theorem 19 will be used in Appendix D to show the following proposition, which shows that the volume-changing behaviors of MWU and OMWU are *opposite* to each other in multi-player game, generalizing a prior result in Cheung & Piliouras (2020).

**Proposition 20.** *The volume integrands of MWU and OMWU in a multi-player game $\mathbf{G}$ are respectively $1 + C_{\mathbf{G}}(\mathbf{p}) \cdot \epsilon^2 + \mathcal{O}(\epsilon^3)$ and $1 - C_{\mathbf{G}}(\mathbf{p}) \cdot \epsilon^2 + \mathcal{O}(\epsilon^3)$. Thus, volume expands locally around a cumulative payoff point $\mathbf{p}$ for MWU (resp. OMWU) if $C_{\mathbf{G}}(\mathbf{p})$ is positive (resp. negative).*

**Multiplayer Potential Game.** By Observation 10, we know that the volume behavior of a potential game is equivalent to a corresponding coordination game in bimatrix game. In this section, we want to show, this holds even in the multi player setting.

**Proposition 21.** *Suppose $\mathcal{P}$ is the potential function of a potential game $\mathbf{U}$. Let $\mathbf{U}^{\mathcal{P}}$ be a game that all players will receive $\mathcal{P}(\mathbf{s})$ when players play strategies $\mathbf{s}$. Then $C_{\mathbf{U}}(\mathbf{p}) = C_{\mathbf{U}^{\mathcal{P}}}(\mathbf{p}) \leq 0$.*

In Appendix E, we will discuss some situations where $C_{\mathbf{U}}(\mathbf{p})$ is strictly less than 0, thus OMWU is Lyapunov chaotic therein.

## D  LOCAL EQUIVALENCE OF VOLUME CHANGE BETWEEN NORMAL-FORM AND GRAPHICAL GAMES

Here, we concern the volume change of a learning algorithm in multi-player game. We first recap from Cheung & Piliouras (2020) on how the volume change is computed for dynamical systems

which are *gradual* (i.e. those governed by a small step-size), followed by a continuous-time analogue of OMWU in games, which are crucial for analyzing the volume change of discrete-time OMWU. Then we compute the volume changes of MWU and OMWU in multi-player graphical games and normal-form games respectively. Once these are done, the proofs of Proposition 20 and Theorem 17 become apparent.

### D.1 DISCRETE-TIME DYNAMICAL SYSTEMS AND VOLUME OF FLOW

We consider discrete-time dynamical systems in $\mathbb{R}^d$. Such a dynamical system is determined recursively by a starting point $\mathbf{s}(0) \in \mathbb{R}^d$ and an update rule of the form $\mathbf{s}(t+1) = G(\mathbf{s}(t))$, for some function $G : \mathbb{R}^d \to \mathbb{R}^d$. Here, we focus on the special case when the update rule is *gradual*, i.e. it is in the form of

$$\mathbf{s}(t+1) = \mathbf{s}(t) + \epsilon \cdot F(\mathbf{s}(t)),$$

where $F : \mathbb{R}^d \to \mathbb{R}^d$ is a smooth function and step-size $\epsilon > 0$. When $F$ and $\epsilon$ are given, the flow of the starting point $\mathbf{s}(0)$ at time $t$, denoted by $\Phi(t, \mathbf{s}(0))$, is simply the point $\mathbf{s}(t)$ generated by the above recursive update rule. Then the flow of a set $S \subset \mathbb{R}^d$ at time $t$, denoted by $\Phi(t, S)$, is the set $\{\Phi(t, \mathbf{s}) \mid \mathbf{s} \in S\}$. Since $F$ does not depend on time $t$, we have the following equality: $\Phi(t_1 + t_2, S) = \Phi(t_2, \Phi(t_1, S))$.

By equipping $\mathbb{R}^d$ with the standard Lebesgue measure, the *volume* of a measurable set $S$, denoted by $\mathsf{vol}(S)$, is simply its measure. Given a bounded and measurable set $S \subset \mathbb{R}^d$, if the discrete flow in one time step maps $S$ to $S' = \Phi(1, S)$ injectively, then by integration by substitution for multivariables,

$$\mathsf{vol}(S') = \int_{\mathbf{s} \in S} \det\left(\mathbf{I} + \epsilon \cdot \mathbf{J}(\mathbf{s})\right) \, \mathrm{d}V, \tag{9}$$

where $\mathbf{I}$ is the identity matrix, and $\mathbf{J}(\mathbf{s})$ is the *Jacobian* matrix defined below:

$$\mathbf{J}(\mathbf{s}) = \begin{bmatrix} \frac{\partial}{\partial s_1}F_1(\mathbf{s}) & \frac{\partial}{\partial s_2}F_1(\mathbf{s}) & \cdots & \frac{\partial}{\partial s_d}F_1(\mathbf{s}) \\ \frac{\partial}{\partial s_1}F_2(\mathbf{s}) & \frac{\partial}{\partial s_2}F_2(\mathbf{s}) & \cdots & \frac{\partial}{\partial s_d}F_2(\mathbf{s}) \\ \vdots & \vdots & \ddots & \vdots \\ \frac{\partial}{\partial s_1}F_d(\mathbf{s}) & \frac{\partial}{\partial s_2}F_d(\mathbf{s}) & \cdots & \frac{\partial}{\partial s_d}F_d(\mathbf{s}) \end{bmatrix}. \tag{10}$$

Clearly, analyzing the determinant in the integrand in Eqn. (9) is crucial in volume analysis; we call it the *volume integrand*. When the determinant is expanded using the Leibniz formula, it becomes a polynomial of $\epsilon$, in the form of $1 + C(\mathbf{s}) \cdot \epsilon^h + \mathcal{O}(\epsilon^{h+1})$ for some integer $h \geq 1$. Thus, when the step-size $\epsilon$ is sufficiently small, the sign of $C(\mathbf{s})$ dictates on whether the volume expands or contracts.

### D.2 CONTINUOUS-TIME ANALOGUE OF OMWU

OMWU does not fall into the category of dynamical systems defined above, since its update rule is in the form of $\mathbf{s}(t+1) = G(\mathbf{s}(t), \mathbf{s}(t-1))$. Fortunately, Cheung and Piliouras Cheung & Piliouras (2020) showed that OMWU can be well-approximated by the *online Euler discretization* of a system of ordinary differential equations (ODE), and thus it can be well-approximated by a dynamical system.

The ODE system is given below. $\mathbf{p}$ is a dual (cumulative payoff) vector variable, $\mathbf{u} : \mathbb{R}^+ \to \mathbb{R}^d$ is the function such that $\mathbf{u}(t)$ gives the *instantaneous payoff* vector at time $t$. We assume that $\mathbf{u}$ is twice differentiable with bounded second-derivatives, and $\dot{\mathbf{u}}$ denotes the time-derivative of $\mathbf{u}$.

$$\dot{\mathbf{p}} = \mathbf{u} + \epsilon \cdot \dot{\mathbf{u}}, \tag{11}$$

*Online Euler discretization* (OED) of Eqn. (11) refers to the following time-discretization of the ODE system. In applications, $\dot{\mathbf{u}}$ might not be explicitly given, and the sequence $\mathbf{u}(0), \mathbf{u}(1), \mathbf{u}(2), \cdots$ are available *online* (i.e., at time $t$ we only have access of $\mathbf{u}(\tau)$ for $\tau = 0, 1, \cdots, t$). As the discretization step is $\epsilon$, we approximate $\dot{\mathbf{u}}(t)$ by $(\mathbf{u}(t) - \mathbf{u}(t-1))/\epsilon$. By using this approximation, OED of Eqn. (11) yields

$$\mathbf{p}(t+1) = \mathbf{p}(t) + \epsilon \cdot \left[\mathbf{u}(t) + \epsilon \cdot \frac{\mathbf{u}(t) - \mathbf{u}(t-1)}{\epsilon}\right] = \mathbf{p}(t) + \epsilon \cdot [2 \cdot \mathbf{u}(t) - \mathbf{u}(t-1)],$$

which is exactly the OMWU update rule in general context.

When compared the OED with the standard Euler discretization

$$\mathbf{p}(t+1) = \mathbf{p}(t) + \epsilon \cdot [\mathbf{u}(t) + \epsilon \cdot \dot{\mathbf{u}}(t)],$$

OED incurs a local error that appears due to the approximation of $\dot{\mathbf{u}}(t)$. The local error can be bounded by $\mathcal{O}(\epsilon^3)$. Cheung and Piliouras Cheung & Piliouras (2020) showed that eventually the determinant of the volume integrand is a of the form $1 + C(\mathbf{s}) \cdot \epsilon^2 + \mathcal{O}(\epsilon^3)$, the local error does not affect the first and second highest-order terms, and hence can be ignored henceforth.

### D.3 MWU in Graphical Games

Let $\mathbf{H}$ be a graphical game of $N$ players, where between every pair of Players $i$ and $k$, the payoff bimatrices are $(\mathbf{H}^{ik}, (\mathbf{H}^{ki})^\mathsf{T})$. In the cumulative payoff space, let $\mathbf{p} = (\mathbf{p}_1, \cdots, \mathbf{p}_N)$ denote the cumulative payoff profile, and let $\mathbf{x} = (\mathbf{x}_1, \cdots, \mathbf{x}_N)$ denote the corresponding mixed strategy profile, where $\mathbf{x}_i$ is a function of $\mathbf{p}_i$. We will write $\mathbf{x}_i$ and $\mathbf{x}_i(\mathbf{p}_i)$ interchangeably. The expected payoff to strategy $j$ of Player $i$ is

$$u_{ij}(\mathbf{p}) = \sum_{\substack{k \in [N] \\ k \neq i}} [\mathbf{H}^{ik} \cdot \mathbf{x}_k(\mathbf{p}_k)]_j,$$

which will be used to compute the Jacobian matrices of MWU and OMWU.

For MWU, the Jacobian matrix $\mathbf{J}$ is a squared matrix with each row and each column indexed by $(i, j)$, where $i$ is a Player and $j \in S_i$. The precise values of its entries are given below:

$$\forall j_1, j_2 \in S_i, \;\; \epsilon J_{(i,j_1),(i,j_2)} = \epsilon \cdot \frac{\partial u_{ij_1}}{\partial p_{ij_2}} = 0 \tag{12}$$

and

$$\forall i \neq k, \; j \in S_i, \; \ell \in S_k, \;\; \epsilon J_{(i,j),(k,\ell)} = \epsilon \cdot \frac{\partial u_{ij}}{p_{k\ell}} = \epsilon x_{k\ell} \cdot \left(H_{j\ell}^{ik} - [\mathbf{H}^{ik} \cdot \mathbf{x}_k]_j\right). \tag{13}$$

Then by expansion using Leibniz formula, the determinant of $(\mathbf{I} + \epsilon \cdot \mathbf{J})$ is

$$1 - \sum_{\substack{i \in [N] \\ j \in S_i}} \sum_{\substack{k > i \\ \ell \in S_k}} (\epsilon J_{(i,j),(k,\ell)})(\epsilon J_{(k,\ell),(i,j)}) + \mathcal{O}(\epsilon^3)$$

$$= 1 - \epsilon^2 \cdot \sum_{\substack{i \in [N] \\ j \in S_i}} \sum_{\substack{k > i \\ \ell \in S_k}} x_{ij} x_{k\ell} \left(H_{\ell j}^{ki} - [\mathbf{H}^{ki} \cdot \mathbf{x}_i]_\ell\right) \left(H_{j\ell}^{ik} - [\mathbf{H}^{ik} \cdot \mathbf{x}_k]_j\right) + \mathcal{O}(\epsilon^3). \tag{14}$$

By noting the similarity of the double summation to $C_{(\mathbf{A},\mathbf{B})}(\cdot)$ in Eqn. (4), we can immediately rewrite the above expression as

$$1 + \epsilon^2 \cdot \sum_{i,k:1 \leq i < k \leq N} C_{(\mathbf{H}^{ik}, (\mathbf{H}^{ki})^\mathsf{T})}(\mathbf{p}_i, \mathbf{p}_k) + \mathcal{O}(\epsilon^3). \tag{15}$$

### D.4 OMWU in Graphical Games

For OMWU, as we pointed out already, we will first consider its continuous analogue first. Thus, we need to compute $\dot{\mathbf{u}}$ in the continuous-time setting. By chain rule, we have

$$\dot{u}_{ij}(\mathbf{p}) = \sum_{\substack{k \in [N] \\ k \neq i \\ \ell \in S_k}} \frac{\partial [\mathbf{H}^{ik} \cdot \mathbf{x}_k(\mathbf{p}_k)]_j}{\partial p_{k\ell}} \cdot \frac{\mathrm{d}p_{k\ell}}{\mathrm{d}t} = \sum_{\substack{k \in [N] \\ k \neq i \\ \ell \in S_k}} x_{k\ell} \cdot \left(H_{j\ell}^{ik} - [\mathbf{H}^{ik} \cdot \mathbf{x}_k]_j\right) \cdot \frac{\mathrm{d}p_{k\ell}}{\mathrm{d}t},$$

and hence

$$\frac{\mathrm{d}p_{ij}}{\mathrm{d}t} = \sum_{\substack{k \in [N] \\ k \neq i}} [\mathbf{H}^{ik} \cdot \mathbf{x}_k]_j + \epsilon \cdot \sum_{\substack{k \in [N] \\ k \neq i \\ \ell \in S_k}} x_{k\ell} \cdot \left(H_{j\ell}^{ik} - [\mathbf{H}^{ik} \cdot \mathbf{x}_k]_j\right) \cdot \frac{\mathrm{d}p_{k\ell}}{\mathrm{d}t}.$$

Note that this is a recurrence formulae for $\frac{d\mathbf{p}}{dt}$. By iterating it[12], we have

$$
\frac{dp_{ij}}{dt} = \sum_{\substack{k \in [N] \\ k \neq i}} [\mathbf{H}^{ik} \cdot \mathbf{x}_k]_j + \epsilon \cdot \sum_{\substack{k \in [N] \\ k \neq i \\ \ell \in S_k}} x_{k\ell} \cdot \left( H_{j\ell}^{ik} - [\mathbf{H}^{ik} \cdot \mathbf{x}_k]_j \right) \cdot \left( \sum_{\substack{r \in [N] \\ r \neq k}} [\mathbf{H}^{kr} \cdot \mathbf{x}_r]_\ell \right) + \mathcal{O}(\epsilon^2).
$$

Hence, its standard Euler discretization, which approximates the OED with local error $\mathcal{O}(\epsilon^3)$, can be written as below (where we ignore the $\mathcal{O}(\epsilon^3)$ error terms):

$$
p_{ij}(t+1) = p_{ij}(t) + \epsilon \sum_{\substack{k \in [N] \\ k \neq i}} [\mathbf{H}^{ik} \cdot \mathbf{x}_k]_j + \epsilon^2 \sum_{\substack{k \in [N] \\ k \neq i \\ \ell \in S_k}} x_{k\ell} \cdot \left( H_{j\ell}^{ik} - [\mathbf{H}^{ik} \cdot \mathbf{x}_k]_j \right) \cdot \left( \sum_{\substack{r \in [N] \\ r \neq k}} [\mathbf{H}^{kr} \cdot \mathbf{x}_r]_\ell \right).
$$

With this, we are ready to compute the Jacobian matrix $\mathbf{J}$ for OMWU. For all $j_1, j_2 \in S_i$,

$$
\epsilon J_{(i,j_1),(i,j_2)} = \epsilon^2 \sum_{\substack{k \in [N] \\ k \neq i \\ \ell \in S_k}} x_{k\ell} \cdot \left( H_{j_1\ell}^{ik} - [\mathbf{H}^{ik} \cdot \mathbf{x}_k]_{j_1} \right) \cdot x_{ij_2} \cdot \left( H_{\ell j_2}^{ki} - [\mathbf{H}^{ki} \cdot \mathbf{x}_i]_\ell \right) \qquad (16)
$$

and for all $i \neq k, j \in S_i, \ell \in S_k$,

$$
\epsilon J_{(i,j),(k,\ell)} = \epsilon x_{k\ell} \left( H_{j\ell}^{ik} - [\mathbf{H}^{ik} \cdot \mathbf{x}_k]_j \right) + \mathcal{O}(\epsilon^2) \qquad (17)
$$

Then by expansion using Leibniz formula, the determinant of $(\mathbf{I} + \epsilon \cdot \mathbf{J})$ is

$$
1 + \left( \underbrace{\sum_{\substack{i \in [N] \\ j \in S_i}} \epsilon J_{(i,j),(i,j)}}_{T_1} - \underbrace{\sum_{\substack{i \in [N] \\ j \in S_i}} \sum_{\substack{k > i \\ \ell \in S_k}} \left( \epsilon J_{(i,j),(k,\ell)} \right) \left( \epsilon J_{(k,\ell),(i,j)} \right)}_{T_2} \right) + \mathcal{O}(\epsilon^3).
$$

By a direct expansions on $T_1$ and $T_2$, it is easy to see that $T_1 = 2T_2$ (after ignoring $\mathcal{O}(\epsilon^3)$ terms). On the other hand, the coefficient of $\epsilon^2$ in $T_2$ is exactly the same as the double summation in Eqn. (14), thus it equals to $-\sum_{i,k:1 \leq i < k \leq N} C_{(\mathbf{H}^{ik}, (\mathbf{H}^{ki})^\top)}(\mathbf{p}_i, \mathbf{p}_k)$. Overall, we show that the determinant equals to

$$
1 - \epsilon^2 \cdot \sum_{i,k:1 \leq i < k \leq N} C_{(\mathbf{H}^{ik}, (\mathbf{H}^{ki})^\top)}(\mathbf{p}_i, \mathbf{p}_k) + \mathcal{O}(\epsilon^3). \qquad (18)
$$

**Observation 22.** *The coefficient of $\epsilon^2$ in Eqn. (18) is the exact negation of the coefficient of $\epsilon^2$ in Eqn. (15).*

### D.5 COMPLETING THE LOCAL EQUIVALENCE PROOF

In a multiplayer normal-form game $\mathbf{G}$, recall that notation Eqn. (1). We point out the following formulae:

$$
\frac{\partial U_{j_1 j_2 \cdots j_g}^{i_1 i_2 \cdots i_g}}{\partial p_{ij}} = 0 \qquad \qquad \text{if } i \in \{i_1, i_2, \cdots, i_g\};
$$

$$
\frac{\partial U_{j_1 j_2 \cdots j_g}^{i_1 i_2 \cdots i_g}}{\partial p_{ij}} = x_{ij} \cdot \left( U_{j_1 j_2 \cdots j_g j}^{i_1 i_2 \cdots i_g i} - U_{j_1 j_2 \cdots j_g}^{i_1 i_2 \cdots i_g} \right) \qquad \text{if } i \notin \{i_1, i_2, \cdots, i_g\}.
$$

**MWU.** Here, MWU update rule is $p_{ij}(t+1) = p_{ij}(t) + \epsilon \cdot U_j^i$. When computing the Jacobian matrix for this update rule using the formulae above, and comparing it with the Jacobian matrix computed in Eqn. (12) and Eqn. (13), it is immediate that they are the same by setting $H_{j\ell}^{ik} = U_{j\ell}^{ik}$. This derives Eqn. (5), and completes the proof of Theorem 19.

---

[12]For the formality on why we can do iterations when $\epsilon$ is sufficiently small, see Cheung & Piliouras (2020).

**OMWU.** As before, we use the continuous analogue and compute $\dot{\mathbf{u}}$. By the chain rule and the above formulae, we have

$$\dot{u}_{ij}(\mathbf{p}) \;=\; \sum_{\substack{k\in[N]\\k\neq i\\\ell\in S_k}} \frac{\partial U_j^i}{\partial p_{k\ell}} \cdot \frac{\mathrm{d}p_{k\ell}}{\mathrm{d}t} \;=\; \sum_{\substack{k\in[N]\\k\neq i\\\ell\in S_k}} x_{k\ell} \cdot \left(U_{j\ell}^{ik} - U_j^i\right) \cdot \frac{\mathrm{d}p_{k\ell}}{\mathrm{d}t}$$

and hence

$$\frac{\mathrm{d}p_{ij}}{\mathrm{d}t} \;=\; U_j^i \;+\; \epsilon\cdot \sum_{\substack{k\in[N]\\k\neq i\\\ell\in S_k}} x_{k\ell} \cdot \left(U_{j\ell}^{ik} - U_j^i\right) \cdot \frac{\mathrm{d}p_{k\ell}}{\mathrm{d}t}.$$

Iterating the above recurrence yields

$$\frac{\mathrm{d}p_{ij}}{\mathrm{d}t} \;=\; U_j^i \;+\; \epsilon\cdot \sum_{\substack{k\in[N]\\k\neq i\\\ell\in S_k}} x_{k\ell} \cdot \left(U_{j\ell}^{ik} - U_j^i\right) \cdot U_\ell^k \;+\; \mathcal{O}(\epsilon^2).$$

Its standard Euler discretization is

$$p_{ij}(t+1) \;=\; p_{ij}(t) + \epsilon\cdot U_j^i \;+\; \epsilon^2\cdot \sum_{\substack{k\in[N]\\k\neq i\\\ell\in S_k}} x_{k\ell} \cdot \left(U_{j\ell}^{ik} - U_j^i\right) \cdot U_\ell^k.$$

Now we compute the Jacobian matrix for this standard Euler discretization. For $j_1, j_2 \in S_i$,

$$\epsilon J_{(i,j_1),(i,j_2)} \;=\; \epsilon^2 \sum_{\substack{k\in[N]\\k\neq i\\\ell\in S_k}} x_{k\ell} \cdot \left(U_{j_1\ell}^{ik} - U_{j_1}^i\right) \cdot x_{ij_2} \cdot \left(U_{\ell j_2}^{ki} - U_\ell^k\right)$$

and for all $i \neq k$, $j \in S_i$, $\ell \in S_k$,

$$\epsilon J_{(i,j),(k,\ell)} \;=\; \epsilon x_{k\ell}\left(U_{j\ell}^{ik} - U_j^i\right) + \mathcal{O}(\epsilon^2).$$

By comparing this computed Jacobian matrix with the Jacobian matrix computed in Eqn. (16) and Eqn. (17), it is immediate to see that their determinants are the same (after ignoring all $\mathcal{O}(\epsilon^3)$ terms) by setting $H_{j\ell}^{ik} = U_{j\ell}^{ik}$. With the result we just derived, together with Observation 22 and Theorem 19, Proposition 20 follows.

## E  MULTI-PLAYER POTENTIAL GAME

*Proof of Proposition 21.* We know that the potential game satisfies the following condition:

$$\mathcal{P}(s_i, s_{-i}) - \mathcal{P}(s_i', s_{-i}) = u_i(s_i, s_{-i}) - u_i(s_{i'}, s_{-i}).$$

Therefore, $u_i(s_i, s_{-i}) = \mathcal{P}(s_i, s_{-i}) + v^i(s_{-i})$. Note that $v^i(s_{-i})$ does not depend on $s_i$, the strategy of player $i$.

By Theorem 19, let $\mathbf{H}(\mathbf{U})$ be the induced graphical game of $\mathbf{U}$ and $\mathbf{H}(\mathbf{U}^{\mathcal{P}})$ be the induced graphical game of $\mathbf{U}^{\mathcal{P}}$. Then,

$$
\begin{aligned}
C_{\mathbf{U}}(\mathbf{p}) &= C_{\mathbf{H}(\mathbf{U})}(\mathbf{p}) && \text{(Theorem 19)}\\
&= \sum_{i,k} C_{(\mathbf{H}(\mathbf{U})^{ik}, (\mathbf{H}(\mathbf{U})^{ki})^{\top})}(\mathbf{p}_i, \mathbf{p}_k) && \text{(By equation 15)}\\
&= \sum_{i,k} C_{(\mathbf{H}(\mathbf{U}^{\mathcal{P}})^{ik}, (\mathbf{H}(\mathbf{U}^{\mathcal{P}})^{ki})^{\top})}(\mathbf{p}_i, \mathbf{p}_k) && \text{(see explanation below)}\\
&= C_{\mathbf{H}(\mathbf{U}^{\mathcal{P}})}(\mathbf{p}) && \text{(By equation 15)}\\
&= C_{\mathbf{U}^{\mathcal{P}}}(\mathbf{p}). && \text{(Theorem 19)}
\end{aligned}
$$

The third equality holds as the difference between $\mathbf{H}(\mathbf{U})^{ik}$ and $\mathbf{H}(\mathbf{U}^{\mathcal{P}})^{ik}$ is a trivial matrix:

$$\mathbf{H}(\mathbf{U})^{ik}_{jl} = \mathbf{U}^{ik}_{jl} = \left(\mathbf{U}^{\mathcal{P}}\right)^{ik}_{jl} + \mathbf{E}_{-(i,k)}\left[v^i(s_{-i})\right] = \mathbf{H}(\mathbf{U}^{\mathcal{P}})^{ik}_{jl} + \mathbf{E}_{-(i,k)}\left[v^i(s_{-i})\right];$$

where [13] $\mathbf{E}_{-(i,k)}\left[v^i(s_{-i})\right]$ doesn't depend on $j$, the strategy of player $i$, and only depends on $l$, the strategy of player $k$. The same argument applies for $(\mathbf{H}(\mathbf{U})^{ki})^\mathsf{T}$ and $(\mathbf{H}(\mathbf{U}^{\mathcal{P}})^{ki})^\mathsf{T}$.

To see $C_\mathbf{U}(\mathbf{p}) \leq 0$, observe that the induced graphical game of $\mathbf{U}^{\mathcal{P}}$ between player $i$ and $k$, $(\mathbf{H}(\mathbf{U}^{\mathcal{P}})^{ik}, (\mathbf{H}(\mathbf{U}^{\mathcal{P}})^{ki})^\mathsf{T})$, is also a bimatrix coordination game, which implies $C_{(\mathbf{H}(\mathbf{U}^{\mathcal{P}})^{ik},(\mathbf{H}(\mathbf{U}^{\mathcal{P}})^{ki})^\mathsf{T})}(\cdot) \leq 0$. As $C_\mathbf{U}(\mathbf{p}) = \sum_{i,k} C_{(\mathbf{H}(\mathbf{U}^{\mathcal{P}})^{ik},(\mathbf{H}(\mathbf{U}^{\mathcal{P}})^{ki})^\mathsf{T})}(\mathbf{p}_i, \mathbf{p}_k)$, the result follows. □

Next, we identify several cases such that $C_\mathbf{U}(\mathbf{p})$ is strictly negative in the region

$$S^\delta = \{\mathbf{x} | \forall i, j \; x_{ij} > \delta\}.$$

The conditions we pose are on the corresponding potential function $\mathcal{P}$. Note that $\mathbf{H}(\mathbf{U}^{\mathcal{P}})^{ik}$, the induced edge-game between player $i$ and $k$, is also a coordination game, i.e. $\mathbf{H}(\mathbf{U}^{\mathcal{P}})^{ik} = (\mathbf{H}(\mathbf{U}^{\mathcal{P}})^{ki})^\mathsf{T}$.

- Case 1:

$$\min_{\mathbf{x},\mathbf{g},\mathbf{h}} \sum_{1 \leq i < k \leq N} \sum_{j \in S_i, \ell \in S_k} \left(P^{ik}_{j\ell} - g^{ik}_j - h^{ik}_\ell\right)^2 \geq \theta,$$

where $P^{ik}_{j\ell} = \mathbb{E}_{\mathbf{s}_{-(i,k)}}\left[\mathcal{P}(s_i = j, s_k = \ell, \mathbf{s}_{-(i,k)})\right]$. With this condition, we can prove that $C_\mathbf{U}(\mathbf{p}) \leq -\theta\delta^2$ for any $\mathbf{p}$ in $S^\delta$. One key observation for this is true is that

$$C_\mathbf{U}(\mathbf{p}) = \sum_{i,k} C_{(\mathbf{H}(\mathbf{U}^{\mathcal{P}})^{ik},(\mathbf{H}(\mathbf{U}^{\mathcal{P}})^{ki})^\mathsf{T})}(\mathbf{p}_i, \mathbf{p}_k)$$

$$= -\sum_{i,k} \sum_{j,\ell} x_{ij}(\mathbf{p}_i) x_{k\ell}(\mathbf{p}_k) \left(P^{ik}_{j\ell} - g^{ik}_j - h^{j\ell}_\ell\right)^2,$$

as $\mathbf{H}(\mathbf{U}^{\mathcal{P}})^{ik} = \mathbf{U}^{\mathcal{P}ik} = \mathbf{P}^{ik}$.

- Case 2:
  If $\mathbf{U}$ is a graphical game, then if there exists a pair of player $i_1$ and $i_2$, such that the game between $i_1$ and $i_2$ is a non-trivial game, then $C_\mathbf{U}$ will be strictly negative in $S^\delta$.

- Case 3:
  Consider the payoff matrix of $\mathbf{U}^{\mathcal{P}}$, the coordination game, between players $i_1$ and $i_2$ given a strategy profile of the other players. There are total $\prod_{i \neq i_1, i_2} n_i$ such matrices, one for each strategy profile of the other players, and each matrix is of dimension $n_{i_1} \times n_{i_2}$. We call these matrices the *projected matrices* for players $i_1, i_2$.
  Let $\mathcal{M}$ denote the matrix space of $n_{i_1} \times n_{i_2}$. On the other hand, trivial matrices form a subspace of dimension $n_{i_1} + n_{i_2} - 1$.[14] Let's call this the *trivial space*, denoted by $\mathcal{T}$.
  We consider the direct decomposition $\mathcal{M} = \mathcal{T} \oplus \mathcal{V}$. Let a set of bases of $\mathcal{M}$ be $\mathbf{B}_1, \mathbf{B}_2, \mathbf{B}_3, \cdots, \mathbf{B}_{n_{i_1} n_{i_2}}$, where the first $n_{i_1} + n_{i_2} - 1$ bases form a basis of $\mathcal{T}$, and the remaining bases form a basis of $\mathcal{V}$. Without loss of generality, we assume that all bases are of L$_2$ norm 1.[15]

---

[13] $\mathbf{E}_{-(i,k)}\left[v^i(s_{-i})\right]$ is the expectation over all the strategies taken by the players other than $i$ and $k$ and $v^i(s_{-i})$ does not depend on the strategy taken by player $i$.

[14] Recall that a trivial matrix $\mathbf{T}$ can be represented as $\{u_j + v_k\}_{j,k}$. Consider the natural linear map $L$ such that $L(u_1, u_2, \cdots, u_{n_{i_1}}, v_1, v_2, \cdots, v_{n_{i_2}})$ maps to the trivial matrix $\mathbf{T}$. Note that the kernel of $L$ is of dimension 1, since if $L(u_1, u_2, \cdots, u_{n_{i_1}}, v_1, v_2, \cdots, v_{n_{i_2}})$ is the zero matrix, then we must have $v_k = -u_j$ for all $j, k$, and hence the kernel of $L$ must be the span of the vector $(\underbrace{1, 1, \cdots, 1}_{\text{the } u \text{ part}}, \underbrace{-1, -1, \cdots, -1}_{\text{the } v \text{ part}})$. Thus, the dimension of all trivial matrices is the dimension of the domain of $L$, which is $n_{i_1} + n_{i_2}$, minus the dimension of the kernel of $L$.

[15] Here, the norm is defined w.r.t. the standard Frobenius matrix inner product.

Given the above-mentioned bases of $\mathcal{M}$, each of the projected matrices can be written into a unique linear combination of these bases. Now, suppose there is a base $\mathbf{B}_\ell$ for $l \geq n_{i_1} + n_{i_2}$ (i.e. this base is in the set of bases for $\mathcal{V}$), such that all projected matrices have non-positive (or non-negative) coefficients of this base, and at least one of these projected matrices (which we call a *special projected matrix*) has strictly negative (or strictly positive) coefficient of the base. Then we claim that $C_{(\mathbf{H}(\mathbf{U}^{\mathcal{P}})^{i_1 i_2}, (\mathbf{H}(\mathbf{U}^{\mathcal{P}})^{i_2 i_1})^\intercal)}(\mathbf{p}_{i_1}, \mathbf{p}_{i_2})$ will be strictly negative in $S^\delta$. This is because $\mathbf{H}(\mathbf{U}^{\mathcal{P}})^{i_1 i_2}$ is a convex combination of all those projected matrices, and by our assumption above, when $\mathbf{H}(\mathbf{U}^{\mathcal{P}})^{i_1 i_2}$ is expressed as the linear combinations of the bases of $\mathcal{M}$, the coefficient of $\mathbf{B}_\ell$ is strictly negative (or strictly positive), thus $\mathbf{H}(\mathbf{U}^{\mathcal{P}})^{i_1 i_2}$ cannot be a trivial matrix.

Suppose further that there exists $\theta > 0$ such that a special projected matrix has negative (or positive) coefficient for $\mathbf{B}_\ell$ which is smaller (or bigger) than $-\theta$ (or $\theta$), then we are guaranteed that $\mathbf{H}(\mathbf{U}^{\mathcal{P}})^{i_1 i_2}$ is bounded away from $\mathcal{T}$ for a distance of $\theta \delta^{N-2}$,[16] and hence as the calculations below show, $C_{(\mathbf{H}(\mathbf{U}^{\mathcal{P}})^{i_1 i_2}, (\mathbf{H}(\mathbf{U}^{\mathcal{P}})^{i_2 i_1})^\intercal)}(\mathbf{p}_{i_1}, \mathbf{p}_{i_2}) \leq -\theta^2 \delta^{2N-2}$. If there exists a pair of player $i_1$ and $i_2$ such that this condition holds, then $C_{\mathbf{U}} \leq -\theta^2 \delta^{2N-2}$.

$$
\begin{aligned}
&C_{(\mathbf{H}(\mathbf{U}^{\mathcal{P}})^{i_1 i_2}, (\mathbf{H}(\mathbf{U}^{\mathcal{P}})^{i_2 i_1})^\intercal)}(\mathbf{p}_{i_1}, \mathbf{p}_{i_2}) \\
&= -\min_{g,h} \sum_{jl} x_{i_1 j}(\mathbf{p}_{i_1}) x_{i_2, l}(\mathbf{p}_{i_2})(\mathbf{H}(\mathbf{U}^{\mathcal{P}})^{i_1 i_2} - g_j - h_k)^2 \\
&\leq -\min_{g,h} \delta^2 \sum_{jl} (\mathbf{H}(\mathbf{U}^{\mathcal{P}})^{i_1 i_2} - g_j - h_k)^2 \\
&= -\delta^2 \sum_{jl} (\mathbf{H}(\mathbf{U}^{\mathcal{P}})^{i_1 i_2} - g_j^* - h_k^*)^2 \\
&\leq -\delta^2 (\theta \delta^{N-2})^2,
\end{aligned}
$$

where $\{g_j^* + h_k^*\}_{jk}$ is projection of $\mathbf{H}(\mathbf{U}^{\mathcal{P}})^{i_1 i_2}$ on the trivial space. The first inequality follows as $\mathbf{p} \in S^\delta$; the second equality holds as the projection minimizing the distance to the trivial space, and the final inequality comes from the distance from $\mathbf{H}(\mathbf{U}^{\mathcal{P}})^{i_1 i_2}$ to the trivial space.

For all these cases, we can have OMWU is $C_{\mathbf{U}}$ to be strictly negative in domain $S^\delta$, which implies OMWU is Lyapunov chaotic in $S^\delta$.

---

[16] To see why, when the coefficient for $\mathbf{B}_\ell$ is bounded away from zero, we are guaranteed that the special projected matrix has a strictly positive distance from $\mathcal{T}$, and this distance is at least $\theta$. Then $\mathbf{H}(\mathbf{U}^{\mathcal{P}})^{i_1 i_2}$, which is a convex combination of all projected matrices where each projected matrix (in particular, the special projected matrix) has a weight at least $\delta^{N-2}$, has a strictly positive distance from $\mathcal{T}$ too, which is at least $\theta \delta^{N-2}$.

