# OpenReview forum: "Chaos of Learning Beyond Zero-sum and Coordination via Game Decompositions"
_ICLR.cc/2021/Conference — ICLR 2021 Poster_

### Official Review · AnonReviewer4 · 2020-10-17
**Some neat tricks but very tough to follow**

**Rating:** 7
**Confidence:** 4

**Review:**

Summary:
This paper studies Lyapunov chaos in learning algorithms for matrix games. It appears to extend earlier work by Cheung and Piliouras to more general-sum settings with the conclusion that in these more common settings the learning algorithms considered exhibit chaos. The paper also presents an interesting notion of matrix domination which is a necessary and sufficient condition for chaos, and also a linear programming approach for the purpose of identifying chaotic games.

Strengths:
* very nice example before section 3.2
* thm 7 is very interesting
* (8) is also very clever

Weaknesses (in rough order of appearance):
* constantly referring to "AI/ML" - just pick one
* "measurement errors in real economies" - why "economies" all of a sudden?
* "Nash equilibrium is not achievable in general." - this is patently false. There is a wealth of recent literature on learning algorithms which stably approach Nash equilibria, see, e.g., https://arxiv.org/pdf/1901.00838
* the whole motivation with roundoff errors reads very naive... these issues have been studied for decades and are taught in standard courses on numerical methods. A good reference which covers these issues with rigor would be the classic book by James Demmel (who is also, coincidentally, known for his work on LAPACK). It is also worth pointing out the extensive work behind (and Turing award for) the development of the IEEE floating point standard. My point being that problems of algorithmic stability arising from finite-precision computation have been (and continue to be) studied extensively, and much theory absolutely considers the presence of computation errors.
* I do not follow footnote 1
* repeated references to "dual space" - what is this? It does not appear to be defined
* (1) is confusing. Some prose description would be good.
* The definitions following "convex regularizer function" are unclear.
* I found (4) onward (through the end of pg. 5) very confusing and didn't really follow
* Notation of "u_i" in Def. 3 is not introduced before
* In 3.2.2 I don't understand why the "iff" property above doesn't apply
* The end at pg. 8 is extremely abrupt. Please offer further interpretation of the results (e.g., what should the reader take away from Thm 11?) and a proper conclusion for the paper.
* [throughout] there are numerous syntax/semantic errors that should be corrected before publication. I suggest employing a copy editor.
* [high level comment] matrix games are a nice theoretical starting point for game theory, but I have yet to see them used in realistic settings. I'm sure that theoretical results may be very interesting, but practically speaking I have difficulty seeing the motivation. I would suggest providing stronger motivation for the reader early on.
* [high level comment] there are *no* experiments at all. Please show at least some numerical examples, if for no other reason than to convince skeptics of the theoretical results.
* [high level comment] numerous times, the paper referred to something like a definition that did not appear until much later in the paper. this is a little awkward for the reader

Overall:
While this paper does offer some interesting ideas, I cannot recommend publication at this time. I have pointed out a number of directions in which the manuscript could be improved; I hope that the authors are able to clarify these points in later revision.

---

> ### Author Response · Authors · 2020-11-13
> **We thank for your detailed review**
>
> Glossary for this reply:
> MWU: Multiplicative Weights Update
> OMWU: Optimistic Multiplicative Weights Update
> ML: Machine Learning
>
> First of all, we thank your detailed review. We are happy to hear that you found "the paper offers some interesting ideas", "Theorem 7 is very interesting" and "the linear program is clever". About the structure of this response, we will first emphasize the key contributions in this paper in the next two paragraphs, followed by addressing the weaknesses you pointed out. We are hoping these can convince you that the paper has good contributions, and the weaknesses are properly fixed or clarified.
>
> Previous works have shown that in some special games or subfamilies of matrix games, learning algorithms (e.g. MWU) are chaotic, e.g. Cheung and Piliouras (COLT 2019 and NeurIPS 2020) theoretically showed that MWU and FTRL in two-person zero-sum games and Optimistic MWU (OMWU) in coordination games are Lyapunov chaotic everywhere in the payoff space. In Learning in Games, we are naturally motivated to understand what happens when MWU or OMWU are applied to *general* matrix games. We vastly extend Cheung and Piliouras' results by characterizing bimatrix games wherein MWU/OMWU is Lyapunov chaotic (almost) everywhere. Such bimatrix games are significantly broader than zero-sum and coordination games.
>
> The above key contribution is achieved via a couple of technical contributions that might inspire future works on chaos analyses of other learning systems. Based upon the volume analyses employed by Cheung and Piliouras, we spot the use of the canonical game decomposition into zero-sum and coordination components, which permit us to analyze the volume-effect of the two components independently. We also introduce a novel matrix domination notion, and the use of a linear program, which facilitate the characterizations.
>
> **Answers to the weaknesses pointed out by Reviewer 4**
>
> >constantly referring to "AI/ML" - just pick one
>
> We were aiming to be more inclusive. We pick ML in the latest revision.
>
> >"measurement errors in real economies" - why "economies" all of a sudden?
>
> We change it to "errors in measuring payoffs".
>
> >"Nash equilibrium is not achievable in general." - this is patently false. There is a wealth of recent literature on learning algorithms which stably approach Nash equilibria
>
> We agree that for zero-sum games, Nash equilibrium is not hard to compute. We are fully aware of the recent stream of "momentum" or "extra-gradient" methods which converge to Nash equilibrium of zero-sum game quickly. By "Nash equilibrium is not achievable in general", we actually mean general matrix games, which are our focus here.
>
> In general, approximating Nash equilibrium is PPAD-hard for matrix games. This means that there should not exist a dynamic which converges to Nash equilibrium quickly (i.e. in poly(\epsilon, #number of players, #number of strategies) time). (https://arxiv.org/abs/1405.3322)
>
> >the whole motivation with roundoff errors reads very naive... these issues have been studied for decades and are taught in standard courses on numerical methods.
>
> First, we thank for your pointers to the classics of Demmel and the great work about algorithmic stability. We would like to clarify the key messages in the paragraph that discusses round-off errors. Our central message is that round-off errors can lead to devastating outcomes, which echoes with the concepts of Lyapunov chaos and unpredictability very well. The target audience of this paragraph is to the broad ML researchers. There is no doubt that researchers working primarily on numerical methods know of the issue very well (so the paragraph might seem naive to them), but we believe the issue does not receive much attention among ML researchers as it should be. To convince ML researchers that this is a legitimate concern, we quoted the example from Ali Rahimi about the learning system, so that ML readers can feel the proximity of the issue. We also relate chaos with the concepts of unpredictability and irreproducibility, which ML researchers are more familiar with.
>
> >I do not follow footnote 1
>
> In Game Theory, different solution concepts arise from various backgrounds. Nash equilibrium is a classical solution concept in Game Theory. From a learning perspective, it is often a fixed point of learning dynamics. But as one of our answers above (the PPAD one) suggest, Nash equilibrium cannot be quickly achieved in general. Thus, researchers have come up with some weaker solution concepts -- coarse correlated equilibrium is one of them. It is known that many online-optimization algorithms are no-regret, and their time averages do converge to a coarsely correlated equilibrium.
>
> In the footnote, we were using the term "no-regret algorithms" instead of "online-optimization algorithms". This seems to make the optimization vs. dynamical-systemic perspectives less transparent, so we have switched to the latter term.
>
> More responses in the next thread...

---

> > ### Author Response · Authors · 2020-11-13
> > **Cont'd**
> >
> > >repeated references to "dual space" - what is this? It does not appear to be defined
> >
> > We apologize for the confusion made. We actually mean the "cumulative payoff space", in which each player updates the cumulative payoffs from each of her strategies.
> >
> > >(1) is confusing. Some prose description would be good.
> >
> > We updated it with a prose description.
> >
> > >The definitions following "convex regularizer function" are unclear.
> >
> > $h_i$ is a (strictly) convex function, which is often called "convex regularizer function". It is used to generate a strategy vector $x^t$ from the cumulative payoff vector $p^t$ via the formula that follows.
> >
> > >I found (4) onward (through the end of pg. 5) very confusing and didn't really follow
> >
> > We apologize for the confusion. Due to the page limit, we just give the definition of $C$ with much explanation, which may be confusing. We have rewritten it in the updated version of the paper. Please have a look and let us know how you think about it.
> >
> > >Notation of "u_i" in Def. 3 is not introduced before
> >
> > $u_i$ is the utility function of player i, which is defined in the paragraph started with "Normal-Form Games" at the start of Section 2.
> >
> > >In 3.2.2 I don't understand why the "iff" property above doesn't apply
> >
> > We are glad to answer this. For one general bimatrix game, it is possible that for most of $j$, $j'$, $k$, $k'$, the matrix of the zero-sum part $\theta$-dominates the coordination part and the $\theta$ is large. However, for one particular $j$, $j'$, $k$, $k'$, zero-sum part is $\theta'$-dominated by the coordination part: $|Z_{jk} + Z_{j'k'} - Z_{jk'} - Z_{j'k}| + \theta' \leq |C_{jk} + C_{j'k'} - C_{jk'} - C_{j'k}|$. Then it certainly violates Definition 5 as the zero-sum part does not always dominate the coordination part. However, all we are interested in is looking at the region, $S^{\delta}$, in which the probability density for any strategy is bounded away from 0. In contrast, the "iff" property is for the whole probability space. Therefore, if $\theta$ is much bigger than $\theta'$, we can infer that $C(\cdot, \cdot)$ is still positive in $S^{\delta}$.
> >
> > >The end at pg. 8 is extremely abrupt. Please offer further interpretation of the results (e.g., what should the reader take away from Thm 11?) and a proper conclusion for the paper.
> >
> > With an extra page given, we are able to add a short conclusion section.
> >
> > >[throughout] there are numerous syntax/semantic errors that should be corrected before publication. I suggest employing a copy editor.
> >
> > Thanks for bringing this up. We have made some updates to the paper in this regard. Please have a look at the updated paper.
> >
> > >[high level comment] matrix games are a nice theoretical starting point for game theory, but I have yet to see them used in realistic settings. I'm sure that theoretical results may be very interesting, but practically speaking I have difficulty seeing the motivation. I would suggest providing stronger motivation for the reader early on.
> >
> > We agree that matrix games are a nice theoretical starting point, since matrix games is among one of the most classical game models, and as our work and many previous works demonstrate, it admits mathematically amenable analyses. It will be interesting to perform stability or chaos analyses on settings that are mathematically more involved but more relevant to applications in ML, e.g., general GANs and differential games. We believe the techniques we use (volume analysis, game decomposition, etc.) can be applied.
> >
> > For the motivation from learning perspective, AnonReviewer3 provides a good answer for us. We surely want to avoid chaos in learning, so by understanding under what situations chaos can arise, we can see when we should adjust or redesign our learning algorithms to counter against chaos.
> >
> > >[high level comment] there are no experiments at all. Please show at least some numerical examples, if for no other reason than to convince skeptics of the theoretical results.
> >
> > Thanks for the suggestion. We have added a graph illumination, that exhibits volume expansion of MWU in a game (which is non-zero-sum and non-coordination) in 2D.
> >
> > >[high level comment] numerous times, the paper referred to something like a definition that did not appear until much later in the paper. this is a little awkward for the reader
> >
> > Thanks for bringing this up. We have caught a few such situations and updated the paper accordingly. Please let us know if we miss anything.
> >
> > Finally, we hope this response can fully address your comments and please let us know if you have any further comments. We sincerely hope that if you find our responses satisfactory, the rating can be adjusted accordingly. Thanks.

---

> > > ### Comment · AnonReviewer4 · 2020-11-19
> > > **Response**
> > >
> > > Thank you for your detailed response and clarification of my many questions. I think I follow all your responses and I will let you know if I have any other questions or suggestions. One very minor suggestion regarding the numerical methods point is that--while I agree with the sentiment that numerical methods do not receive enough attention in the ML community--you tone down this language to say something to that effect (rather than asserting that *all* theory is non-numerical).
> > >
> > > I will update my score accordingly, and look forward to discussion with other reviewers and the editor.

---

> > > > ### Author Response · Authors · 2020-11-22
> > > > **Thanks for your suggestion.**
> > > >
> > > > Thanks for your suggestion. We have updated the paper accordingly, by pointing out that significant efforts have been spent in analyzing and minimizing round-off effects of floating-point computations (see Demmel (1997)). Since evaluations that involve irrational numbers are necessary, round-offs are inevitable, and thus chaotic learning might lead to diverging outcomes and thus unpredictable and irreproducible learning systems.

---

### Official Review · AnonReviewer2 · 2020-10-23
**ICLR 2021 Conference Paper2387 AnonReviewer2**

**Rating:** 7
**Confidence:** 3

**Review:**

Summary:

This paper studies the chaos phenomena of learning in general normal-form games beyond zero-sum and coordination games. Building upon the previous works by Cheung & Piliouras, the authors apply the canonical decomposition of a general bimatrix game to a sum of a zero-sum game and a coordination game. The authors further devise two new techniques: matrix domination and linear program to help analyze the game dynamics.

Detailed Comments:

This paper is generally well-written and clear. The topic concerning the dynamics of learning algorithms is interesting and important for the DL community. This work extends the previous results for zero-sum and coordination games to more general games in a non-trivial way, and it demonstrates the existence of a broad class of games suffering from the chaos phenomena. The ideas of trivial matrices and matrix dominations look nice.

1. It would be better to explain the logic behind C_{(A, B)} more intuitively, since most analysis in this paper focuses on this quantity. It is very interesting that MWU and OMWU have totally opposite dependency on this quantity. The reviewer understands that this result is from the previous work, but it would be better to explain it more clearly and intuitively in this paper to make it self-contained.

2. Could you comment on the games whose dynamic cannot be determined by matrix domination or linear program? For the games that could appear in the DL applications, e.g., GAN, could those games' dynamic be determined by matrix domination or linear program? If not, shall we expect them to have chaos phenomena or not?

---

> ### Author Response · Authors · 2020-11-13
> **We appreciate you taking the time to review our paper.**
>
> We appreciate you taking the time to review our paper. We hope the following response can fully address your comments. Please let us know if you have any further comments.
>
> >It would be better to explain the logic behind $C_{(\mathbf{A},\mathbf{B})}$ more intuitively, ......."
>
> We agree that it is better to explain the logic behind $C_{(\mathbf{A},\mathbf{B})}$ more intuitively. Intuitively, $C_{(\mathbf{A},\mathbf{B})}$ can be viewed as the strength of volume expansion for MWU dynamics in the bimatrix game $(\mathbf{A},\mathbf{B})$. For two-player zero-sum game $(\mathbf{A},\mathbf{-A})$, $C_{(\mathbf{A},\mathbf{-A})}$ is always positive, which means that the MWU dynamics will always expand the volume in the cumulative payoff space. The more positive the $C_{(\mathbf{A},\mathbf{B})}$ is, the quicker the expansion is. In contrast, for two-player coordination game $(\mathbf{A},\mathbf{A})$, $C_{(\mathbf{A},\mathbf{A})}$ is always negative, which means the volume will contract when using the MWU dynamics.
>
> In this paper, we prove that the $C_{(\mathbf{A},\mathbf{B})}$ in general two-player game is the summation of C(the zero-sum part of the game) and C(the coordination part of the game). Intuitively, if the absolute value of C of the zero-sum part is bigger than the absolute value of C of the coordination part, then the value of C of the general game $(\mathbf{A},\mathbf{B})$ will be positive, and this means the MWU dynamic will expand the volume in this game. We'd like to refer you to the two recent papers of Cheung and Piliouras (COLT 2019 and NeurIPS 2020) for more technical details.
>
> Due to the page limit, it is unfortunately hard to include much of the intuitions above, as we already spent 5.2 pages explaining the background materials, and trying to make sure the paper is as self-contained as possible and can be accessible to the broad ML community.
>
> >Could you comment on the games whose dynamic cannot be determined by matrix domination or linear program?
>
> For games which cannot be characterized by matrix domination or linear program, the volume-changing behavior is not that clear-cut. MWU dynamics will expand volume in some regions of the cumulative payoff space, but it will contract volume in the remaining regions. We raise a simple example. Consider a 4x4 bimatrix game, where the top-left 2x2 subgame is a zero-sum, the bottom-right 2x2 subgame is a coordination game, and the top-right and bottom-left subgames are zero games (payoff zero to both players). If both players choose their first two strategies with high probability, then volume expands around that point, as the players are essentially playing a zero-sum game. Analogously, if both players choose their last two strategies with high probability, then volume contracts around that point.
>
> In general, given the zero-sum component (Z,-Z) and coordination component (C,C) of a general game, it is possible to identify regions wherein volume expands. To see why, note that volume expands in the region {(p,q) | C_{(Z,-Z)}(p,q) + C_{(C,C)}(p,q) > 0}. The condition can be simplified to C_{(Z,-Z)}(p,q) - C_{(C,-C)}(p,q) > 0, which allows us to use Observation 8 to find out these (p,q). In some cases, it is possible to identify a large subset of this region, so we can conclude that chaos occurs in this large subset. (An example is generalized Rock-Paper-Scissors games, see Section 7 of Cheung and Piliouras COLT 2019 paper arxiv version at https://arxiv.org/pdf/1905.08396.pdf.)
>
> >For the games that could appear in the DL applications, e.g., GAN, could those games' dynamic be determined by matrix domination or linear program? If not, shall we expect them to have chaos phenomena or not?
>
> "Does chaos occur in GAN?" is an interesting open problem. AnonReviewer3 has pointed out a reference (UMass Thesis 2018), which has some empirical evidence that chaos can occur (at least locally) with GAN. It is usually easy to show that Lyapunov chaos occurs locally (which boils down to analyzing the eigenvalues of Jacobian at a given point). We believe it will be interesting to see if chaos can occur in a widespread domain, just as what we showed in the cumulative payoff domain for a large family of matrix games.
>
> About the usefulness of game decomposition, matrix domination or linear program in showing chaos in other games, we frankly do not have a good answer at this moment. But this is definitely a good future direction to look into. To shed light on this, we will first need to know better of the mathematical structure of the other games. We believe that for differential games and multi-player games, the game decompositions proposed in Candogan et al. (Flows and decompositions of games: Harmonic and potential games., MOR 2011) and Letcher et al. (Differentiable game mechanics., JMLR 2019) might play a role in chaos analyses. For other games like GANs, as we are not familiar with its mathematical structure, we dare not to comment.

---

### Official Review · AnonReviewer3 · 2020-10-27
**This paper provides tools for classifying the payoff dynamics in general-sum games as Lyapunov chaotic under common FTRL algorithms.**

**Rating:** 7
**Confidence:** 4

**Review:**

This paper provides tools for classifying the payoff dynamics in general-sum n-player games as Lyapunov chaotic assuming three common algorithms are used: multiplicative weights update (FTRL with entropy regularizer), optimistic MWU, and FTRL with L2 regularizer. Previous work (Cheung & Piliouras) showed that the existence of Lyapunov chaos in the dual space is indicated by the sign of a function C of the game. This work shows that this function can be decomposed into a sum: C of the zero-sum part + C of the coordination part. They also show how to use trivial matrices (which don't affect C) to further reduce parts of the game in a way that eases the analysis. As part of their analysis, they prove that the set of bi-matrix games exhibiting chaos has positive Lebesgue measure and discuss how the relative strength of the zero-sum and coordination parts determines the ultimate sign of C.

Overall, I think this is a solid paper and should be accepted. The work is original and significant. The paper could benefit from additional discussion and explanation of the claims (see below) so that the paper is more self-contained.

Definition of Chaos and Interpretation of Results in Primal Space:
Wouldn't the system $\frac{dx}{dt} = x$ with $x(t) = e^t$ satisfy Def 1? In other words, any $\dot{x} = Ax$ with $A \succ 0$ Is this better characterized as divergent? Shouldn't it be required that $\phi(t,B)$ remains inside $\mathcal{O}$ for all time? By inspection, the Lorenz system seems to satisfy this. The authors state "eigenvalues of the local Jacobian characterizes chaotic behaviors". To my knowledge, no linear dynamical system would be said to exhibit chaos. Maybe a sentence or two early on can put to rest a reader's confusion if they already have some background in nonlinear dynamics and chaos theory.

The authors prove chaos in the dual space (which is unbounded), but do not formally relate this to chaos in the primal space (which is bounded). It is stated, "This can be shown to imply instability in the mixed strategy (primal) space." Can the authors provide some explanation or intuition so this part of the paper is more self-contained? It seems like "chaos" according to Def 1 in the dual space is not too surprising because we're measuring divergence of a cumulating function (which naturally defines growth).

Why does the sign of C matter? Can you explain where this Eqn comes from? Can you provide any intuition or must we read the Cheung & Piliouras (2020)? Similar to above, it would be best if this part of the paper was more self-contained.


Quality:
The work is of high quality. I did not check the proofs in the appendix, but those in the main body look correct (Lemma 4 is correct).

Clarity:
The paper is mostly clear, however, I have some questions regarding the definition of chaos (see above) and how to interpret chaos in the dual space in terms of the primal space (which I think we care about more).

Originality:
This work is substantially original. Showing C can be decomposed into a sum of the zero-sum part and coordination part is novel. Also, the tools for removing trivial matrices and bounding C using the output of a linear program is new (assuming we can identify the right trivial matrix). Lastly, proving chaos is "common" is original.

Significance:
Proving chaos is common in games is significant because it means we have to adjust our algorithm design to account for it. Providing tools for determining whether a game exhibits chaos for popular algorithms can tell us whether to redesign the algorithm for that game or to adjust the game in some way.

Minor:

I would suggest backing off the statement that "Understanding chaos is an important but long-ignored aspect of AI/ML". Maybe say it has not received as much attention as it should. A few works come to mind. No need to cite necessarily.
- A RECURRENT NEURAL NETWORK WITHOUT CHAOS (ICLR 2017).
- Exponential expressivity in deep neural networks through transient chaos (NIPS 2016).
- FROM OPTIMIZATION TO EQUILIBRATION: UNDERSTANDING AN EMERGING PARADIGM IN ARTIFICIAL INTELLIGENCE AND MACHINE LEARNING (UMass Thesis 2018, Section 5.7); approximates the Lyapunov exponents of GANs to identify chaos.

"We answer the above question affirmatively". Instead of "How does...", say "Does the..." so that the question admits a yes/no response.

"–The second technique" - remove the dash.

"of a multi-player potential games" - game singular.

"mixed strategy space of dimension n" - the simplex defined with n-probabilities is (n-1)-dimensional.

Explain $\mathbb{E}[A_{jk}] = \mathbb{E}[A_j] = \mathbb{E}[A_k]$. This is not obvious to me, and in fact, appears trivially false so I'm probably misunderstanding your notation. I can easily construct an A matrix and mixed strategy y such that this does not hold.

Maybe say $u=v=[2, 0, 2]$ to construct the trivial matrix in the Simple Example.

Use logical and "$\land$" maybe instead of $\texttt{and}$ in Section 3.2.

Def 4 looks like the magnitude of the determinant of the elementwise log of a matrix. I just thought I'd share. I'm not sure if there's anything there, but determinant pops up in volume analysis as you know and the matrices can represent exponential expansion so the log maybe makes sense by a very hand-wavy connection.

"it is easy to see that there is quite many games" - are quite many

"zero-sum part $\theta$-dominate" - dominates

Can you provide a constructive way to discover the trivial matrix defined by $g_j$ and $h_k$ in Eqn 8? Or in general?

---

> ### Author Response · Authors · 2020-11-13
> **We thank for your detailed and supportive review**
>
> First of all, we thank for your detailed and supportive review. In particular, we thank for the pointers to works about chaos in ML, which do provide us some useful materials to look into for future work. We address your comments as below.
>
> >Definition of Chaos and Interpretation of Results in Primal Space: Wouldn't the system $dx/dt = x$ with $x(t) = e^t$ satisfy Def 1? In other words, any $dx/dt = Ax$ with $A\succ 0$ is this better characterized as divergent?
>
> First, about linear dynamical system, you are definitely right that $A\succ 0$ implies diverging trajectories, but by convention they are usually not said to be chaotic, since they admit "closed-form" solution. Having closed-form solution is very rare for non-linear dynamical systems though.
>
> >Shouldn't it be required that $\phi(t,B)$ remains inside $\mathcal{O}$ for all time? By inspection, the Lorenz system seems to satisfy this. The authors state "eigenvalues of the local Jacobian characterizes chaotic behaviors". To my knowledge, no linear dynamical system would be said to exhibit chaos. Maybe a sentence or two early on can put to rest a reader's confusion if they already have some background in nonlinear dynamics and chaos theory.
>
> About "$\phi(t,B)$ should remains inside $\mathcal{O}$ for all time", you are right that it is the standard way for definition. Cheung and Piliouras had implicitly fine-tuned the standard definition to fit the special need for their analyses. The reason for this fine-tuning is because when $\mathcal{O}$ is unbounded, the volume-changing behavior in a region that is far away from the starting point can be expanding but with diminishing rate. Thus, to show that the volume is increasing substantially and exponentially fast, we have to confine the definition of chaos in a bounded set --- this is how the definition of "Lyapunov chaotic everywhere" in Def 1 is made. But when $\mathcal{O}$ is bounded, the exponential volume expansion clearly cannot go on forever, and that's why in Def 1 the standard condition "$\phi(t,B)$ should remains inside $\mathcal{O}$ for all time" is removed.
>
> The fine-tuned definition of Lyapunov chaos can be used to derive some surprising negative properties of MWU in zero-sum games (see Cheung and Piliouras NeurIPS 2020 paper), so we believe it is a notion of Lyapunov chaos that worth pursuing in the context of learning in games.
>
> >The authors prove chaos in the dual space (which is unbounded), but do not formally relate this to chaos in the primal space (which is bounded). It is stated, "This can be shown to imply instability in the mixed strategy (primal) space." Can the authors provide some explanation or intuition so this part of the paper is more self-contained? It seems like "chaos" according to Def 1 in the dual space is not too surprising because we're measuring divergence of a cumulating function (which naturally defines growth).
>
> We are happy to answer this. We agree that the divergence of the cumulating functions is not too surprising. Yet, the divergence in Def 1 is *exponential*, which is far from a triviality.
>
> For the formal relation between chaos in dual space and instability in primal space, Cheung and Piliouras (NeurIPS 2020) have made this in their Proposition 2 (page 5 of https://arxiv.org/pdf/2005.13996.pdf). Intuitively, if all players use the mixed strategies in $S^{\delta}$, which the probability of every strategy is at least $\delta$ (i.e. the probability is bounded away from zero), then we can see the cumulative payoffs of all strategies grow at most linearly, which implies the growth of the volume in the dual space is at most polynomial. So, if the dual set corresponding to $S^{\delta}$ has exponential volume expansion, then a set that lies in $S^{\delta}$ initially will finally escape $S^{\delta}$. Further discussions on the relation between dual and primal spaces can be found in Appendix A of their paper (Cheung and Piliouras (NeurIPS 2020)).
>
> >Why does the sign of C matter? Can you explain where this Eqn comes from?......
>
> The function C was derived by Cheung and Piliouras (COLT 2019). We gave an explanation in page 5: the sign (of C) determines the local volume-changing behavior around the point (p, q) when MWU is used. Thus, when C(p,q) is positive, the MWU dynamic around the point (p,q) is volume-expanding, while negative C(p,q) implies volume contraction around (p,q). We have added a footnote to explain it more thoroughly.
>
> >I would suggest backing off the statement that "Understanding chaos is an important but long-ignored aspect of AI/ML". Maybe say it has not received as much attention as it should.
>
> >We answer the above question affirmatively". Instead of "How does...", say "Does the..." so that the question admits a yes/no response.
>
> Thanks! We have followed your suggestion.
>
> More responses in the next thread...

---

> > ### Author Response · Authors · 2020-11-13
> > **Cont'd**
> >
> > > "–The second technique" - remove the dash.
> >
> > >of a multi-player potential games" - game singular.
> >
> > > mixed strategy space of dimension n" - the simplex defined with n-probabilities is (n-1)-dimensional.
> >
> > Fixed.
> >
> > > Explain E[A_jk] = E[A_j] = E[A_k]......
> >
> > We apologize for missing the definitions of $A_j$ and $A_k$. We have added them back in page 5 of the updated paper. Their definitions are: $A_j = \sum_k A_{jk} y_k$ and $A_k = \sum_j A_{jk} x_j$.
> > Therefore, $E[A_{jk}] = \sum_{jk} x_j A_{jk} y_k = \sum_{j} x_j \sum_k A_{jk} y_k = \sum_j x_j A_j = E[A_j]$; this also applies to $E[A_k]$.
> >
> > > Maybe say u = v = [2, 0 ,2] to construct the trivial matrix in the Simple Example.
> >
> > > Use logical ∧ maybe instead of and in Section 3.2.
> >
> > Thanks! We have followed your suggestion.
> >
> > > Def 4 looks like the magnitude of the determinant of the elementwise log of a matrix. I just thought I'd share. I'm not sure if there's anything there, but determinant pops up in volume analysis as you know and the matrices can represent exponential expansion so the log maybe makes sense by a very hand-wavy connection.
> >
> > This connection looks interesting. Indeed the arithmetic in the volume analysis of MWU in game is beautiful, which makes me suspect that there might be more elegant way to explain.
> >
> > However, to our knowledge, we do not see how to relate Def 4 to the magnitude of the determinant of the elementwise log of a matrix. It will be great if you can provide us a pointer (say, a book title, or the name of a researcher that has works related to this notion).
> >
> > > it is easy to see that there is quite many games" - are quite many......
> >
> > > "zero-sum part -dominate" - dominates"
> >
> > Fixed. Thanks!
> >
> > > Can you provide a constructive way to discover the trivial matrix defined by $g_j$ and $h_k$ in Eqn 8? Or in general?
> >
> > The Eqn 8 is a linear program. In general, it can be solved by ellipsoid method. There might be more "elementary" way to solve the linear program, but we don't know (yet).
> >
> > We hope these answers can address your comments. Please let us know if you have any further comments.

---

> > > ### Comment · AnonReviewer3 · 2020-11-19
> > > **Request Additional Explanation of Basic Concepts Included in Main Body**
> > >
> > > Regarding the boundedness of $\mathcal{O}$. Thanks for your explanation. I originally overlooked the refinement after Def. 1 that states "A d.s. is Lyapunov chaotic everywhere if it is Lyapunov chaotic in any ***bounded*** open set.". Is there any reason you cannot add the word "bounded" to the definition statement as well? i.e., "A d.s. is Lyapunov chaotic in an open ***bounded*** set $\mathcal{O}$..."
> > >
> > > Thank you for explaining the relation between chaos in the dual space and instability in the primal space. To be clear, I am specifically asking you to include this kind of explanation in the main body of the paper. This submission already relies heavily on the Cheung and Piliouras (Neurips 2020) work and it would help the paper's readability if a high level summary of this relevant intuition was included so that this paper can better stand on its own.
> > >
> > > Regarding $C$, I appreciate the high level explanation of the sign of $C$ denoting expansion or contraction, but I am more asking what does $C$ denote mathematically? Is $C$ related to the determinant of the Jacobian of the dynamics? From skimming Cheung & Piliouras (2019), it looks like this is in fact the case. I would add a sentence in the paragraph after Proposition 1 saying something like "it was shown in Cheung & Piliouras (2019, 2020) that the volume changing behavior is given by $1 + C \cdot \epsilon^2 + \mathcal{O}(\epsilon^4)$. Therefore, if $C > 0$, then the volume is increasing indicating separation of trajectories, i.e., chaos." Without this, $C$ just appears as this magical constant connected to volume in some unknown way.
> > >
> > > It looks like $A_j = \nabla_x [x^\top A y]$ and $A_k = \nabla_y [x^\top A y]$. It might be helpful to identify this for reader given the paper is analyzing the learning dynamics, which are directly controlled by the gradients.
> > >
> > > I haven't thought much about the determinant of the elementwise-log of the matrix and I don't know of any references looking at this. It was more just food for thought.
> > >
> > > I see that you now define $g_j^*$ and $h_k^*$ at the end of appendix B. I did not see them defined anywhere near Equation 8 in the original manuscript. Thanks for clearing this up.

---

> > > > ### Author Response · Authors · 2020-11-22
> > > > **Thanks for your comments.**
> > > >
> > > > Thanks for your comments. We hope the following responses can answer your questions.
> > > >
> > > > > Regarding the boundedness of $\mathcal{O}$
> > > >
> > > > First, recall that $\mathcal{O}$ is a set in the cumulative payoff space. In Def. 1, there are indeed two definitions. The first definition is "Lyapunov chaotic in an open set $\mathcal{O}$". We actually use this definition for unbounded $\mathcal{O}$. To be specific, in Theorems 9 and 11, we show that those systems are Lyapunov chaotic in $S^\delta$ (defined in page 7), where
> > > > $S^\delta =$ {$ (\mathbf{p},\mathbf{q}) | \forall j\in S_1, k\in S_2, x_j(\mathbf{p}) \ge \delta \wedge y_k(\mathbf{q}) \geq \delta $},
> > > > which is an unbounded subset in the cumulative payoff space. To see why it is unbounded, observe that if $(\mathbf{p},\mathbf{q})$ is in $S^\delta$,
> > > > then for any real numbers $c_1,c_2$ and for the all-one vector $\mathbb{1}$, $(\mathbf{p} + c_1 \cdot \mathbb{1},\mathbf{q}+c_2 \cdot \mathbb{1})$ is also in $S^\delta$.
> > > >
> > > > The second definition is "Lyapunov chaotic everywhere", which means the system is Lyapunov chaotic in any *bounded* open set. This boundedness condition is a simple sufficient (but not necessary) condition to guarantee that for MWU in non-trivial zero-sum game, the volume is increasing substantially and exponentially fast within the set. Actually, we do not need this definition for presenting our results, but we did use it for stating the results of Cheung and Piliouras in the introduction. This is why we still include it.
> > > >
> > > > It seems that having two definitions in the same Def. environment is slightly confusing. So we now put them in separate Def. environments.
> > > >
> > > >
> > > > >I am specifically asking you to include this kind of explanation in the main body of the paper.
> > > >
> > > > Thanks. We now included one paragraph after Corollary 3 which gives the intuitions behind the proof of this corollary.
> > > >
> > > > >Regarding $C$.
> > > >
> > > > Thank you for your suggestion. Yes, $C$ is related to the determinant of the Jacobian of the dynamics. We add one paragraph in the introduction, the second paragraph of "Our Contributions", to explain the relationship between *C* and the volume analysis. We also add your suggested sentence as a footnote (due the page limit) at the bottom of page 5.
> > > >
> > > > >It looks like $A_j = \nabla_{x_j} [x^\top A y]$ and $A_k = \nabla_{y_k} [x^\top A y]$.
> > > >
> > > > We agree that your suggestion and we add your definition. We also keep our elementary definition, $A_j = \sum_k A_{jk} y_k$, which shows a strong relationship between $A_j$ and the payoff of the strategy $j$. For those dynamics such as MWU and OMWU, they are more directly related to the payoffs. The elementary definition also enables audience to check our proofs more easily.
> > > >
> > > > Thanks for your comments again. Please let us know if you have any further comments.

---

> > > > > ### Comment · AnonReviewer3 · 2020-11-24
> > > > > **The paper looks good to me - Please just address my linear program comment below**
> > > > >
> > > > > Thanks for your explanations and your changes to the paper. I think it reads much more clearly now. I have one more comment regarding the linear program formulation, another regarding Def 1, and then a few more regarding minor suggestions / typos.
> > > > >
> > > > > Equation 7 appears incomplete. I'm not sure if you accidentally reverted some of your changes, but $g$ and $h$ are no longer defined. As it is currently written, the linear program in (7) is only given $K$ as input. And $r$ is the variable to be optimized over. For this problem to be well defined, $g_j$ and $h_k$ must either be passed as input or defined in terms of $K$ and $r$. The following sentence "Note that $(g_j + h_k)_{j,k}$ constructs a trivial matrix" does not clear this up because no direct formula for constructing a trivial matrix is given.
> > > > >
> > > > > Regarding our discussion of Def 1. I understand that this is the current working definition of chaos in the related work and so I understand sticking with this, but I would like to see future work try to adhere to a more stringent requirement for chaos that distinguishes between "divergent", "sensitivity to initial conditions", and "chaotic". Going back to the linear dynamical system example I gave before, $\frac{dx}{dt} = e^{x(t)}$ would be deemed "chaotic" according to Def 1 in the open set $t \in (-1, 1)$ with Lyapunov exponent $\gamma=1$, $\lambda=1$ and clearly, this system is simply divergent. Chaos has many definitions, but usually includes a few more properties than just a) sensitivity to initial conditions. Unless I am mistaken, Def 1 does not include any notion of b) periodic orbits or c) mixing / coverage of the open set. These additional properties are critical to defining "chaos" as something interesting. Divergence is quite a simple phenomenon by comparison. That being said, I understand the authors state there is a relationship between the dual and primal spaces that is addressed in previous work that says the volume expansion in the dual (cumulative payoff) space implies a more standard phenomenon of chaos in the primal (strategy) space.
> > > > >
> > > > > In the paragraph below “Our Contributions”
> > > > > - "all agents in a bimatrix game G uses" -- should be use
> > > > > - "when a set S..." -- maybe add "set S in the payoff space" to be clear
> > > > >
> > > > > In the paragraph after "Our Contributions"
> > > > > - "canonical decomposition of such game" -- should be games plural
> > > > >
> > > > > In the paragraph before related work:
> > > > > - "are opposite to each other" needs a period
> > > > >
> > > > > In Section 2
> > > > > - "We let −(i1, · · · , ig) denotes" -- should be denote
> > > > >
> > > > > Below Proposition 3:
> > > > > - "analyzing the sign the function" -- sign of the function
> > > > > - "dynamic system" -- dynamical?
> > > > >
> > > > > Below Observation 10:
> > > > > - "volume-changing behavior of potential game" -- a potential game
> > > > >
> > > > > Simple example:
> > > > > - "who chooses strategy" -- a strategy (occurs in two places)
> > > > >
> > > > > Section 3.2.2, 2nd paragraph:
> > > > > - "This is not the only case we can bound" -- where we can bound
> > > > > - "Even the entries in matrix" -- Even if the

---

> > > > > > ### Author Response · Authors · 2020-11-24
> > > > > > **Thanks for your detailed comments.**
> > > > > >
> > > > > > Thanks for your detailed comments. We have fixed the grammatical errors and typos you pointed out.
> > > > > >
> > > > > > > Equation 7 appears incomplete. I'm not sure if you accidentally reverted some of your changes, but g and h are no longer defined.
> > > > > >
> > > > > > For the linear program, we apologize for the lack of clarity. $g_j$ and $h_k$ are the auxiliary variables used to optimize *r*. They are not input or defined in terms of *K*. The solution to this linear program will also output $g_j$ and $h_k$, which is a trivial matrix. In the paper, we now add $\mathbf{g}$ and $\mathbf{h}$ as a subscript of $\min$ to indicate they are the variables in this linear program.
> > > > > >
> > > > > > > I would like to see future work try to adhere to a more stringent requirement for chaos that distinguishes between "divergent", "sensitivity to initial conditions", and "chaotic". Going back to the linear dynamical system example I gave before (...) Chaos has many definitions, but usually includes a few more properties than just a) sensitivity to initial conditions. Unless I am mistaken, Def 1 does not include any notion of b) periodic orbits or c) mixing / coverage of the open set. These additional properties are critical to defining "chaos" as something interesting.
> > > > > >
> > > > > > You are right that there are other chaos notions which capture other properties meaning "unstable" at a high level, including the two properties you mentioned: "periodic orbits" and  "mixing / coverage of the open set". For example, Li-Yorke chaos is capturing the property there are infinitely pairs of trajectories which get close and then move away repeatedly, and one canonical path to proving Li-Yorke chaos is to prove there is a period-three point. (see https://en.wikipedia.org/wiki/James_A._Yorke for more discussions about Li-Yorke chaos.)
> > > > > >
> > > > > > Having this said, we agree that the classical definitions of Lyapunov chaos do care about divergence but not periodicity or mixing/coverage. It is very interesting to see whether the learning-in-game systems we study satisfy other chaos notion or not. But frankly it might not be easy; as far as I know, many Li-Yorke chaos results are established to one-dimensional systems or very specific higher-dimensional systems only.
> > > > > >
> > > > > > About what we can do: We add a footnote after Def. 1 saying some linear dynamical systems are considered Lyapunov chaotic although they can have simple closed-form solutions. But for nonlinear dynamical systems, which mostly do not admit closed-form solutions, Lyapunov chaos is well-received as a notion that captures unpredictability. In Related Work section we recently add, we give some pointers to recent Li-Yorke chaos results about learning-in-game. When the space is more adequate in the future version of this paper, we can add more discussions about other chaos notions.

---

> > > > > > > ### Comment · AnonReviewer3 · 2020-11-24
> > > > > > > **Thank you for making changes!**
> > > > > > >
> > > > > > > Footnote 7 and the update to equation (7) clear things up for me. Thank you for accommodating my nitpicking.

---

### Official Review · AnonReviewer1 · 2020-10-28
**Weak presentation that harms understanding of the results and contribution**

**Rating:** 5
**Confidence:** 3

**Review:**

As far as I understand, the authors study Lyapunov chaos that occur in learning in non-zero-sum games.


Main issues:
1.	The current structure of the paper is very unusual what makes the presentation of the key results very unclear. Just for instance:
-	Given overall 8 pages, the presentation of claimed results starts from 6th page (before that we have Introduction and Preliminaries)
-	No conclusion: the main text chops on Theorem 11 without any discussion on this theorem after.
2.	Continuing presentation issues, I did not get at all the core of the contributions. I’ve expected to get them in the paragraph titled “Our Contributions.” (in Page 2). But this and next paragraphs provide no any insight on the contributions of this paper.
3.	It would be interesting to get insights on how the paper is related to learning community: despite the authors articulate on this in Abstract and Introduction, it is still unclear how the obtained results can improve learning / learning algorithms and/or improve their application in practice.

I believe that even notable and important contributions for the community must have a significantly better presentation, because otherwise it will be quite difficult for readers to obtain and understand these results.


Minor issues:
-	Abstract is too long for this ICLR format.
-	I’m not sure that the format of ICLR allows have Appendix in the main text instead of using supplementary materials.
=====


I raise up my score after the authors' changes

---

> ### Author Response · Authors · 2020-11-13
> **Thanks for taking time to review this paper**
>
> Glossary for this response:
> MWU: Multiplicative Weights Update
> OMWU: Optimistic Multiplicative Weights Update
> FTRL: Follow-The-Regularized-Leader algorithm
> ML: Machine Learning
>
> Thank you for taking the time to review this paper. Our responses to all the comments are as follows.
>
> **(1) Composition of the paper (5 pages of intro+prelim, 3 pages of technical contents)**
>
> We realized this is an unusual composition, but it is necessary for the following reason. This paper really requires a lot of background materials from Learning in Games, Dynamical Systems, Volume Analysis, and also some results from the two recent papers of Cheung and Piliouras (COLT 2019 and NeurIPS 2020). These materials are still new to most ML researchers. To make sure the paper is as self-contained as possible and can be accessible to the broad ML community, we made the difficult decision to spend five pages on intro+prelim. Even after this, we still had some materials which we could only shallowly cover due to page limit; for them, we give pointers to Cheung and Piliouras, which provide more elaborate discussions.
>
> We understand some reviewers and readers are more eager to grasp the main contributions of our work, so we devoted roughly one page (page 3) early in the paper to provide layman discussion about our main contributions. We will elaborate on this in Part (2) of this response.
>
> As an additional page is allowed, in the updated version of the paper, we add a conclusion section and enrich the paper with some experiments/graphical illuminations. This should make the paper composition more balanced. The conclusion section includes a discussion on the significance of this paper to the learning community and possible future directions, which we also discuss in Part (3) of this response.
>
> **(2) "Our Contributions"**
>
> We devoted one page (page 3) to provide layman discussions about our main contributions. "Our Contributions" serves as a roadmap to understanding our results, which are stated from page 6 onwards. We admit that the insights are not there yet, but we did provide them at the beginning of Section 3.2, which needs a key technical Lemma 4. This is why the insights appear so late. To help with grasping our main contributions, we would like to provide a concise extract of "Our Contributions":
>
> 1. Previous Work We Base On: Our work is based on the previous two papers of Cheung and Piliouras (COLT 2019 and NeurIPS 2020), which showed that MWU and FTRL in two-person zero-sum games and Optimistic MWU (OMWU) in coordination games are Lyapunov chaotic everywhere in the payoff space. To show this, they use volume analysis. In brief, given an open ball of initial conditions, which has a positive Lebesgue measure (volume), we analyze how the volume of the ball changes when it is evolved by the learning process. Cheung and Piliouras reduced volume analysis of MWU in general bimatrix game $(\mathbf{A},\mathbf{B})$ to analyzing the sign of a function $C_{(\mathbf{A},\mathbf{B})}(*)$, which is a technical starting point of our work.
> 2. Motivation: The previous papers of Cheung and Piliouras focus on special sub-families of games. But in Learning in Games, we are naturally motivated to understand what happens when MWU or OMWU are applied to *general* matrix games.
> 3. Insights: We provided some insights at the beginning of Section 3.2. By Lemma 4, we can decompose the function $C_{(\mathbf{A},\mathbf{B})}(*)$ neatly into zero-sum and coordination parts, which correspond to the strengths of volume-expansion and volume-contraction. Informally, when the zero-sum part is "larger" than the coordination part, we have volume-expansion (and hence chaos) almost everywhere. The comparison of "largeness" of the two parts are made via the notion of matrix domination (Section 3.2.1) and the use of a linear program (Section 3.2.2).
> 4. Techniques/Tools: We brought up the main tools (the direct-sum decomposition of a game into zero-sum and coordination components, trivial matrices, matrix domination, a linear program) we use in the later stage of the paper. Their precise definitions and discussions appear from page 6 onwards.
>
> **(3) How is the paper related to the learning community?**
>
> First of all, AnonReviewer3 provides a good answer for us. We surely want to avoid chaos in learning, so by understanding under what situations chaos can arise, we see when we should adjust or redesign our algorithm to counter against chaos.
>
> Studying learning in matrix (normal-form) games is a good theoretical starting point, since matrix game is among one of the most classical game models, and as our work and many previous works demonstrate, it admits mathematically amenable analyses. Having the above said, we are immensely interested in pursuing volume and chaos analyses on settings that are more relevant to applications in ML, e.g., general GANs and differential games. We believe the techniques we use (volume analysis, game decomposition, etc.) can be applied.

---

> > ### Author Response · Authors · 2020-11-13
> > **Cont'd**
> >
> > **(4) Conclusion and Others**
> >
> > After the above replies, we hope you can understand why we choose an unusual but necessary composition of the paper, and our intended purpose of the "Our Contributions" section. We provide a concise extract of "Our Contributions", which we hope can facilitate your understanding of our contributions better. We are happy to address any further concerns about the presentation.
> >
> > Q: Abstract is too long for this ICLR format.
> >
> > A: We have removed some less essential parts from the abstract.
> >
> > Q: I’m not sure that the format of ICLR allows have Appendix in the main text instead of using supplementary materials.
> >
> > A: From ICLR Call for Papers: "Authors may use as many pages of appendices (after the bibliography) as they wish, but reviewers are not required to read these." We believe this implies we can attach the appendix after the main body and the bibliography in the same submission file.
> >
> > We hope this response can address your concerns. Please let us know if you have any further comments or concerns.

---

> > > ### Comment · AnonReviewer1 · 2020-11-20
> > > **Some suggestions**
> > >
> > > Thank you for your reply and the update of the paper. The presentation looks better.
> > >
> > > However, I believe that the presentation can be improved still. Below are my arguments and suggestions. Please, consider them not as arguments for a rejection rather than points where the work could be improved to be more popular and more visible for a larger group of readers.
> > >
> > >
> > > I understand that there are topics that needs to be introduced heavily to readers from ML community. And it is important to provide a clear setup, definitions, references and literature review. But “clear” does not mean more words. I believe that these things can be made briefly and at the same time succinctly.
> > >
> > > In particular, this means that the introduction can be significantly reduced, saving space for more preliminaries, discussions and literature review. I believe that with small efforts we can reduce Introduction from 2.5 pages to 1 page without reducing the level of understandings at all. Just because the whole sufficient information (even for non-familiar readers) starts from Preliminaries. Thus, Introduction plays a role of a concise «squeeze» of prelims and results. But in your case, on the one hand, it is larger than Prelim + Conclusions; and, on the other hand, it contains too lot informal sentences that put together a reader into frustration.
> > >
> > > Below are some suggestions:
> > > 1.	I like the informal introduction itself, but it is good for a journal submission rather than for 8(9)-page work. I believe that it is easy to reduce Intro by consolidating into 2 paragraphs all the paragraphs that are situated from the beginning to “Our Contributions. Our starting point is the…”
> > > 2.	At least we can:
> > > -	Unite the 2nd and 3rd paragraphs and state the same ideas using 2-3 sentences.
> > > -	Significantly reduce the paragraph starting from “To see the importance for the ML community to understand chaos, recall that one of our primary..”: We can just left 2-3 key ideas here and put more reference for those who are not familiar for further reading (it is a standard practice: otherwise we will spend paragraphs for every widely known information) – Note that AnonReviewer4 indicated excessiveness of this part as well
> > > -	(using the strategy from the previous point) Significantly reduce the paragraph starting from “A natural and generic approach to extend those..”
> > > 3.	You say that pg.3 is layman, but a layman will fail understand anything when reaching the paragraph “The volume-expansion argument crucially…” because it contains a mysterious function C. A text for a layman must contain only well-known notions, but C is not known even for specialists (who not read [Cheung & Piliouras (2019; 2020). ) So, **please**, if it is very difficult to remove this, make at least the following changes:
> > > -	“analyzing a function” -> “analyzing the function”
> > > -	“denotes the underlying game” -> “denotes an underlying game”
> > > -	put citation Cheung & Piliouras (2019; 2020).  right after “C_G(\cdot)” to indicate that this notion is introduced there.
> > >
> > > Note that other reviewers indicated the issue with point 3, and I believe it is not fixed for now.

---

> > > > ### Author Response · Authors · 2020-11-22
> > > > **Thanks for your suggestions.**
> > > >
> > > > Thanks for your constructive suggestions. We agree that they can improve our work to be more popular and more visible for a larger group of readers. Accordingly, we significantly condense the introduction. We unite the 2nd and 3rd paragraphs into one. After this, we give the motivation of our work, and a discussion of the main results in Cheung and Piliouras (COLT2019, NeurIPS2020). We move the discussion of game decomposition techniques into the related work section.
> > > >
> > > > In "Our Contributions", we agree that the paragraph starting with "the volume-expansion argument crucially..." is not easy to understand by general audience. To make this part more accessible, we add one paragraph to explain the relationship between the function $C$ and the volume analysis. With this paragraph, we believe $C$ will no longer be a mysterious function. We have also replaced some technical statements of our contributions with more comprehensible statements.
> > > >
> > > > We also add a related work section to the main body. We give a literature review of some recent works on using dynamical system tools to tackle ML-related problems, and discuss game decomposition techniques.
> > > >
> > > > Thanks again for your suggestions. Please let us know if you have any further comments.

---

### Decision · Program_Chairs · 2021-01-07
**Final Decision**

**Decision:**

Accept (Poster)

**Comment:**

This paper looks at chaos in learning in games, extending a line of work in two players zero-sum games (that I found quite restrictive in the past). It somehow reduces the class of more general games to zero-sum and cooperative games (this decomposition is already known) so that the techniques can be transposed here.

The paper is interesting, yet sometimes difficult to follow, and I am not certain that it gives many new insights.

Nonetheless, we believe its quality justify acceptance.